# Assimilation of river discharge in a land surface model to improve estimates of the continental water cycles

Fuxing WANG[1], Jan POLCHER[1], Philippe PEYLIN[2], and Vladislav BASTRIKOV[2]

[1]Laboratoire de Météorologie Dynamique, IPSL, CNRS, Ecole Polytechnique, 91128, Palaiseau, France

[2]Laboratoire des sciences du climat et de l'environnement, IPSL, CEA, Orme des Merisiers, 91191, Gif sur Yvette, France

Manuscript revised on June 12, 2018

To be submitted to *Hydrology and Earth System Sciences* (*HESS*)

[*]Correspondence to:

Fuxing Wang

Email: fuxing.wang@lmd.jussieu.fr

Tel: 0033 (0)1 69 33 51 80

**Abstract:**

The river discharge plays an important role in earth's water cycle, but it is difficult to estimate due to un-gauged rivers, human activities, and measurement errors. One approach is based on the observed flux and a simple annual water balance model (ignoring human processes) for ungauged rivers, but it only provides annual mean values which is insufficient for oceanic modellings. Another way is by forcing a land surface model (LSM) with atmospheric conditions. It provides daily values but with uncertainties associated to models.

We use data assimilation techniques by merging the modelled river discharges by ORCHIDEE (without human processes currently) LSM and the observations from Global Runoff Data Center (GRDC) to obtain optimized discharges over the entire basin. The 'model systematic errors' and 'human impacts' (e.g., dam operation, irrigation, etc.) are taken into account by an optimization parameter $x$ (with annual variation), which is applied to correct model intermediate variables runoff and drainage over each sub-watershed. The method is illustrated over the Iberian Peninsula with 27 GRDC stations over the period 1979-1989. ORCHIDEE represents a realistic discharge over north of the Iberian Peninsula with small model systematic errors, while the model overestimates discharges by 30%-150% over south and northeast region where the blue water footprint is large. The normalized bias has been significantly reduced to less than 30% after assimilation, and the assimilation result is not sensitive to assimilation strategies. This method also corrects the discharge bias for the basins without observations assimilated by extrapolating the correction from adjacent basins. The 'correction' increases the inter-annual variability of river discharge because of the fluctuation of water usage. The $E$ ($P$-$E$) of GLEAM (Global Land Evaporation Amsterdam Model, v3.1a) is lower (higher) than the bias corrected value, which could be due to the different $P$ forcing and probably the missing processes in the GLEAM model.

Key words: river discharge; data assimilation; human processes; water cycle; land surface model; the Mediterranean

## 1. Introduction

The river discharge is an essential component of the earth's water cycles, which can be used as an indicator of the hydrological cycle intensification (Munier et al., 2012). It is important not only for water resources management, climate studies, ecosystem health over land (Syed et al., 2010; Sichangi et al, 2016), but also for providing freshwater inflow to ocean (Dai and Trenberth, 2002). The freshwater flux at the sea surface has significant influence on the climate system (e.g., ENSO, ocean dynamics) and on ocean salinity (Kang et al., 2017). The fresh water inputs for ocean model usually requires high frequency data (e.g., daily or 10-daily, Scherbakov and Malakhova 2011). Besides, as the ocean model with high spatial resolution (e.g., < 10 km) demonstrates better skills than coarse resolution model (Bricheno et al., 2014; Wang et al., 2017), there is also a requirement of high resolution fresh water fluxes. Although great efforts have been made for gridded river discharge data at global scale (e.g., RivDIS v1.1, Vorosmarty et al., 1998; Dai and Trenberth, 2002; Fekete et al., 2002), these data are usually at monthly or annual scales and have not been updated with time. Therefore, it is of great interest to estimate large scale river discharge over the long-term at high temporal and spatial resolution and low uncertainty.

Estimating the river discharge input to ocean is a difficult endeavor for several reasons. First, there are many un-gauged rivers that are difficult to evaluate. Second, most large rivers are gauged by national agencies, and these data are difficult to access for public users. Besides, the number of operational gauging stations is decreasing worldwide (Syed et al., 2010; Sichangi et al, 2016). Third, even though the observations are available, the observed river flow at the outlet is not well known because it is difficult to get gauging stations close to the river mouth and many observations are affected by human activities especially in semi-arid regions (Jordà et al., 2017).

One approach to estimate the freshwater inflow into ocean is based on the observed water fluxes over data-rich regions and a simple annual water balance model, precipitation inputs minus the evaporation, which ignoring human usage and other processes over ungauged basins (e.g., Szczypta et al. 2012; Peucker-Ehrenbrink, 2009; Mariotti et al., 2002; Struglia et al. 2004; Boukthir and Barnier, 2000; Ludwig et al., 2009). This method is the basis of most water balance studies and oceanic modelling activities but it has several limitations. First, there are uncertainties in observations related to measurement method and post-processing method. These uncertainties are

difficult to quantify due to the incomplete information (Jordà et al., 2017). Second, only annual
mean values are available over un-gauged basins (about 40% for the Mediterranean; 42% over
globe excluding Greenland and Antarctica, Clark et al., 2015) by simple runoff models, which are
not sufficient for oceanic modellings.

Riverine input can also be obtained through forcing a state of the art land surface model
(LSM) or global hydrological model (GHM) with bias corrected atmospheric conditions (e.g., aus
der Beek et al., 2012; Bouraoui et al. 2010; Jin et al., 2010; Sevault et al., 2014). These numerical
models can estimate river discharge at higher frequency and over more un-gauged basins (Jordà et
al., 2017), but they are associated with modelling uncertainties. First, models are designed and
have proved the ability to capture the natural water cycles, but relatively less progress has been
made in parameterizing human processes (Pokhrel et al., 2017). The water flow of many
catchments has been strongly regulated by human through irrigation use, dam operation, etc. (e.g.,
the southern shores of the Mediterranean). Second, there are large discrepancies among models
resulting from the differences in model inputs, parameterizations, and atmospheric forcing data
(Ngo-Duc et al., 2007; Wang et al., 2016; Liu et al. 2017).

The objective of the present study is to illustrate a novel approach based on assimilation
techniques applied to LSM to estimate continental water cycles (riverine fresh water). The data
assimilation, a specific type of inverse problem, is generally applied for different cases: (1) to
correct initial condition (correcting state variable) which is mostly used for numerical weather
prediction; (2) to correct the state variable during the data assimilation period (i.e., in this case
both the trajectory of the model and the initial conditions are corrected); and (3) to correct the
parameter of a model by optimization. In the current study, the data assimilation refers to the 3rd
case. This assimilation approach merges the data from the model (ORCHIDEE LSM) and the
observed river discharge from the Global Runoff Data Centre (GRDC, 56068 Koblenz, Germany).
This will allow to compensate for model systematic errors or missing processes and provide
estimates of the riverine input into the sea at high temporal and spatial resolution. Although
previous works exist on assimilation of river discharge (e.g., Li et al., 2015; Bauer-Gottwein et al.,
2015; Pauwels et al., 2009), these studies mainly focus on the stream flow prediction over
individual catchments. They are difficult to extend to long-term scale and large catchment due to
the observations and computing time limitations.

This paper focuses on the methodology and its illustration in a Mediterranean region (the Iberian Peninsula) which is considered one of the most vulnerable regions to climate change due to its geographic and socio-economic characteristics (Vargas-Amelin and Pindado, 2014). Although the amount of river discharge is relatively small (about one third to half of precipitation amount; Tixeront, 1970; Shaltout and Omstedt 2015), it is an important source of fresh water entering the Mediterranean Sea and it plays an important role in sustaining the marine productivity (Bouraoui et al., 2010) and overturning circulation (Verri et al., 2017). The river discharges to the Mediterranean Sea underwent important changes during recent decades. This variation is particularly important for this region because of its scarce water resource with increasing water demand for domestic, industrial, irrigation and tourism activities, as well as its drier and warmer conditions under climate change (Romanou et al., 2010). Considering the high stress on the water resources in the Mediterranean region, accurate estimation of the actual resources is important.

The methods (including the model, datasets and numerical experiment) are described in Sect. 2. The results and discussions are given in Sect. 3. Conclusions are drawn in Sect. 4.

## 2. Methods

### 2.1. The theoretical background

The theoretical basis of the LSM assimilation for the study is the vertical and lateral water balance. The precipitation ($P$) input of a basin is transferred into either evaporation, surface runoff ($R$), deep drainage ($D$) (eventually the $R$ and $D$ reaching the channel and leaving in the form of river discharge), or stored in the ground.

$$\frac{dW}{dt} = P - (R + D) - E,  \tag{1}$$

Over long period, the change of water storage $\frac{dW}{dt}$ is small ($\frac{dW}{dt} \approx 0$), thus

$$P - E \approx R + D  \tag{2}$$

The lateral water balance over a basin (e.g., the sub-catchment 2 in blue in Fig. 1a) is given by:

$$\frac{dA_2}{dt} = \left[ \iint_{S_2} (R_2 + D_2)\, ds \right] - Q_2 + Q_1, \qquad (3)$$
where $S_2$ is the area of sub-catchment 2; $A_2$ is the water stored in the aquifers of area $S_2$; $Q_2$ and $Q_1$
are the river discharge at outlet of each sub-catchment, and they are calculated by the integral of
runoff and drainage over the sub-catchment area $S_1$ and $S_2$. We assume the $A_2$ variation at annual
scale is small ($\frac{dA_2}{dt} \approx 0$) due to its slow variability, although it can be nonzero due to the human
intervention (e.g., over Indo-Gangetic Basin, MacDonald et al., 2016). The $W$ and $A$ terms refer to
water storage and water stored in the aquifers, respectively. The Eqs. (1)-(3) describe the basic
water cycle processes in the LSMs.
Despite that the LSMs have developed rapidly during the last few decades, few models
take into account the human water usage processes. Due to this limitation, LSMs are usually
accompanied with errors in reproducing discharge and evaporation in areas where these processes
are dominant. Assuming the $P$ forcing is known in LSM, the modelled water continuity imposes a
balance of errors between $E$, $R$ and $D$. However, the $R$ and $D$ are conceptual variables, and their
errors are impossible to evaluate by observations directly. The field measurements of $E$ over large
area are also scarce due to land surface heterogeneity (Kalma et al., 2008). Fortunately, the
observations of river discharge ($Q_{obs}$) are available. By fitting modelled discharge with $Q_{obs}$, we
can correct model intermediate variables in Eqs. (1)-(3) (e.g., correct $R$ and $D$ by a correction factor
$x$, Fig. 1a) in order to get bias corrected river discharge ($Q_{corr}$).
$$Q_{corr} = \int_{catchment} (x \cdot R + x \cdot D)\, dS, \qquad (4)$$
Recalling the $\frac{dW}{dt}$ is small and $P$ is known, we then transfer the $x$ into vertical water balance
and close the horizontal water balance by the corrected evaporation ($E_{corr}$):
$$E_{corr} \approx P - x \cdot (R + D), \qquad (5)$$
The impacts of assimilation on $E$ ($\Delta E$) can be derived from the optimal $x$, $R$, and $D$:
$$\Delta E = E_{corr} - E \approx (1 - x) \cdot (R + D), \qquad (6)$$

The key problem remains to determine the optimal $x$ (described in Sect. 2.2.2). Each

discharge observation station corresponds to an optimal correction factor $x$ since the discharge is
the only representative of the integral over the basin. The total number of $x$ depends on the number
of available stations. The optimal $x$ over each observation station is applied to its entire upstream
area. Over each upstream area (dashed box in Fig. 1a), the optimal $x$ of these model grid cells are
the same. The '$R + D$' and $E$ are corrected at the same grid cell level by $x$ and Eq. (5), respectively.
**2.2. The models**
**2.2.1. Assimilation strategy and ORCHIDAS**

The optimal $x$ is obtained from the ORCHIDEE Data Assimilation System (ORCHIDAS,

https://orchidas.lsce.ipsl.fr/). It was designed to optimize the variables related to water, energy and
carbon cycles in ORCHIDEE (Organising Carbon and Hydrology in Dynamic Ecosystems;
Krinner et al. 2005; De Rosnay et al., 2002) LSM by using various observations (e.g. in situ,
satellite, etc.). The ORCHIDAS has been applied over different regions for various variables and
demonstrated good performance (Santaren et al., 2007; Kuppel et al., 2012; MacBean et al., 2015).
More details of ORCHIDAS are presented by Peylin et al. (2016).

In this work, the ORCHIDAS drives the ORCHIDEE routing scheme which is

computationally less expensive than the full ORCHIDEE model (Fig. 1b). The data assimilation
approach relies on the minimization of a misfit function $J(x)$ (aka cost function) by successive calls
to "gradient-descent" minimization algorithm L-BFGS-B (Limited-memory Broyden-Fletcher-
Goldfarb-Shanno algorithm with simple Box constraints, Byrd et al., 1995).

A new vector of parameter values $x$ is estimated at each iteration. The $J(\mathbf{x})$ measures the

mismatch between the vector of observed river discharges $Q_{obs}$ and corresponding simulated
values $Q_{sim}(x)$, as well as between the optimized correction factors $x$ and its prior information $x_{prior}$:
$$J(\mathbf{x}) = [\mathbf{Q}_{obs} - \mathbf{Q}_{sim}(\mathbf{x})]^t \mathbf{R}^{-1} [\mathbf{Q}_{obs} - \mathbf{Q}_{sim}(\mathbf{x})] + (\mathbf{x} - \mathbf{x}_{prior})^t \mathbf{B}^{-1} (\mathbf{x} - \mathbf{x}_{prior}), \qquad (7)$$
where $\mathbf{R}$ and $\mathbf{B}$ represent the prior error covariance matrices for observations and parameters,
respectively. Diagonal elements of $\mathbf{R}$ matrix represent the data uncertainties, which include both
the measurement errors (systematic and random) and model errors, we have defined it as the root
mean squared error (RMSE) between the prior model simulations and the observed river
discharges. Non-diagonal elements describe correlations between the data, which however are
difficult to presume correctly, and are usually neglected. The prior parameter uncertainties (matrix
**B**) have been set to 40% of the range of variation of correction factors obtained from the ratio $Q_{obs}$
and first guess value of river discharge simulation ($Q_{fg}$) obtained from $x_{prior}$. The matrix **B** was
determined based on the expert knowledge of ORCHIDEE model (Kuppel et al., 2012; Santaren
et al., 2014). Correlations between prior parameter values have not been considered. The gradient
of the $J(x)$ is calculated for all the parameters by finite difference approach at each iteration
(Kuppel et al., 2012).

## 2.2.2. ORCHIDEE LSM with high-resolution river routing model

The ORCHIDEE LSM is the land component of Institut Pierre Simon Laplace Climate

Model (IPSL-CM), which simulates energy, water and carbon cycles between the soil and
atmosphere. The unsaturated water flow is described at each land point by the one-dimensional
Richards equation with 2 m soil discretized to 11 levels. The surface runoff and deep drainage at
bottom layer are computed by Horton overland flow and free drainage (equals to hydraulic
conductivity), respectively. In other words, the ORCHIDEE LSM assumes that the aquifer level
is below the model bottom, and it neglects the upward water flow through capillary forces from its
underlying aquifer. The evaporation is partitioned into transpiration, bare soil evaporation,
interception loss and snow sublimation.

The ORCHIDEE is coupled with the ocean model through the river routing scheme

(Polcher, 2003; Ducharne et al. 2003; Guimberteau et al., 2012) which computes river discharge
by integrating the surface runoff and deep drainage over the basin. A high-resolution river routing
scheme was developed recently, which allows to better describe of catchments boundaries, flow
direction, and water residence time (Nguyen-Quang et al., 2018; Zhou et al., 2018). It is based on
the HydroSHED (Hydrological data and maps based on SHuttle Elevation Derivatives at
multiple Scales; http://www.hydrosheds.org/; Lehner et al., 2008) map with 1 km spatial resolution.
There are several hydrological transfer units (HTUs) in one ORCHIDEE grid-cell (e.g., 100 in the
current study). The HTU is constructed based on the Pfafstetter topological coding system and
user defined size. Each HTU represents the section of the river basin within the grid box, and many
HTUs forms a river basin (Nguyen-Quang et al., 2018). Therefore, the relative locations of HTUs
in each grid cell are not fixed.
In each HTU, the water is routed through a cascade of three linear reservoirs characterized
by their residence times: the groundwater, overland and stream reservoirs. The runoff and drainage
are the inputs into the overland reservoir and groundwater reservoir, then they flowed into the
stream reservoir of the downstream sub-grid basin. The residence times are determined by
multiplying a constant reservoir factor ($g$) with a slope index ($k$). The $g$ for stream, overland and
groundwater reservoirs are 0.24, 3, and 25 day/km, respectively (Ngo-Duc et al., 2007). The slope
index is a function of distance ($d$) and slope ($S$) between a pixel and its downstream pixel ($k=d/S^{1/2}$
defined by Ducharne et al., 2003). The water can flow either to the next HTU within the same grid
cell or to the neighboring cell. The river discharge is diagnosed at the HTU level in the assimilation.
The river discharge is linear with $R$ and $D$ at annual scale over a small basin. In case of more than
one observation stations are assimilated in a river basin (e.g., $x_1$ and $x_2$ in Fig. 1a), the river
discharge at downstream is affected by the discharge of upstream thus it is not a linear system
anymore. Therefore, the optimization is needed to deal with the $x$ over the non-linear sub-basins.
The time steps for the ORCHIDEE model and routing scheme are 30 minutes and 3 hours,
respectively. The spatial resolution of the model depends on the resolution of the atmospheric
forcing, and it is 0.5° for the current study (given in Sect. 2.3.2). The soil texture map is from
United States Department of Agriculture (USDA) with 12 soil textures (Reynolds et al. 2000). The
vegetation map is from the European Space Agency Climate Change Initiative (ESA CCI,
https://www.esa-landcover-cci.org/) reduced to the 13 plant functional types represented by the
model.
**2.3. The study domain and the datasets**
**2.3.1. Study domain**
The assimilation system is applied over the Iberian Peninsula. This region is dominated by
two climate types: the oceanic climate in the Atlantic coastal region and the Mediterranean
climate over most of Portugal and Spain. The annual precipitation is extremely unevenly

distributed with more than 1500 mm over northeastern Portugal, much of coastal Galicia and along the southern borders of the Pyrenees but less than 300 mm over southeast Spain (Estrela et al., 2012). Over Spain, agriculture occupies approximately 50% of the land area (e.g., year 2014, https://data.worldbank.org/indicator/AG.LND.AGRI.ZS), and with around 1200 large dams (European Working Group on Dams and Floods, 2010).

## 2.3.2. The meteorology forcing

In order to study the sensitivity of the optimization results to different forcing data, three meteorology forcing are used: WFDEI_GPCC, WFDEI_CRU and CRU_NCEP. The WFDEI_GPCC and WFDEI_CRU (3-hourly, 0.5°) are based on the WFDEI meteorological forcing data which was produced using WATCH (WATer and global CHange) Forcing Data (WFD) methodology applied to ERA-Interim data at 0.5° (Weedon et al., 2014; http://www.eu-watch.org/data_availability). The WFDEI is from 1979 and updates until now with eight meteorological variables at 3-hourly time steps. The precipitation of WFDEI_GPCC and WFDEI_CRU is corrected by GPCC (Global Precipitation Climatology Centre) and CRU (Climatic Research Unit), respectively. The CRU_NCEP (6-hourly, 0.5°) combines the CRU TS.3.1 (0.5°, monthly) climatology covering 1901-2012 and the NCEP (National Centers for Environmental Prediction) reanalysis (2.5°, 6-hour) beginning in 1948 (https://vesg.ipsl.upmc.fr/thredds/fileServer/store/p529viov/cruncep/readme.html). The precipitation of the three forcing is compared with the IB02 which is a gridded daily rainfall dataset for the Iberia Peninsula with 0.2° resolution covers 1950 to 2003 (Belo-Pereira et al., 2011). It is generated by using ordinary kriging from more than 2400 quality-controlled stations.

## 2.3.3. The GRDC dataset

The Global Runoff Database collects the monthly river discharge from most basin agencies around the world (more than 9,300 stations) with an average record length of 43 years. Although the quality of the observations is unknown (e.g., monitoring the river transect, velocity measurements, etc.), the GRDC datasets are the most complete river discharge dataset available today. It is hosted by the German Federal Institute of Hydrology

(Bundesanstalt für Gewässerkunde or BfG;
www.bafg.de/GRDC/EN/Home/homepage_node.html).

**2.3.4. Integration of GRDC in ORCHIDEE**

The location of some stations in the GRDC dataset might be incorrect for either the

longitude or latitude coordinate due to simple typos, logical errors in the original coordinates, or a
swapped order of the coordinate digits (Lehner, 2012). Due to this uncertainty, a quality control is
applied for GRDC when matching it with the corresponding HTUs in the river routing model. For
each GRDC station, the corresponding catchment surface in the model is estimated. The matching
process is stringent, and the GRDC qualification is restricted by two matching criteria: (1) the
difference in upstream area between GRDC and the model is less than a pre-defined percentage;
(2) the distance between GRDC and the model is less than a pre-defined distance. The higher the
two thresholds are, the more the matched GRDC stations can be positioned on the model's basin
representation. Meanwhile, the high threshold increases the uncertainties of the GRDC data due to
the errors in location and upstream area. By compromising between the two contradictory
requirements (the number of GRDC stations and the precise of the data), we choose the threshold
for upstream area difference and distance to be 10% and 25 km, respectively. Under this constraint,
27 GRDC stations are qualified among all 65 stations over the Iberian Peninsula domain (10ºW-
5.5ºE, 34ºN-45.5ºN; Fig. 2). It should be noted one GRDC station can match with several model
HTUs that locate in different model grids. In this case, the HTU with the lowest upstream area
difference is chosen. Therefore, the GRDC station is not necessarily in the same model grid as the
model HTU.

**2.3.5. The evaporation products**

The bias corrected evaporation deduced from the assimilation is compared with the

GLEAM (Global Land Evaporation Amsterdam Model; Martens et al., 2017;
https://www.gleam.eu/) product. GLEAM provides daily evaporation from 1984 to 2011 at 0.25°.
The evaporation is estimated by a minimalistic Priestley-Taylor potential evaporation model with
the majority of inputs estimated from remote sensing. It uses the microwave-derived soil moisture,
land surface temperature and vegetation density, and the detailed estimation of rainfall interception
loss. The rainfall interception loss is estimated separately using the Gash analytical model which
considers the canopy storage capacity, coverage, and the ratio of mean evaporation rate from wet
canopy. There are several versions of GLEAM data available, and we choose the latest version
v3.1a. The precipitation forcing of GLEAM v3.1a is from the Multi-Source Weighted-Ensemble
Precipitation (v1.2).

### 297    2.4. Experiments design

An ORCHIDEE simulation is performed to obtain the $Q_{fg}$ and the corresponding $R$ and $D$.

The ORCHIDAS with L-BFGS-B algorithm explores the full space of $x$ by perturbing a separate
$x$ ($x_i$) over the $i$ th upstream catchment ($i=1, 2, …, N_{opt}$; $N_{opt}$ is the total number of optimized $x$
depending on the number of observation stations) in each iteration. To save computing time, the
river routing parameterization (forced by corrected $R$ and $D$) rather than the full ORCHIDEE is
executed. The total execution time depends on the number of parameters to be optimized, the
length of simulation years, and the number of iterations. Multi-level parallelisms of the
assimilation are implemented to achieve the high computational efficiency. In each iteration, the
assimilation can run with $N_{opt}$ 'river routing' simulations, with each 'river routing' model
parallelized with $N_{routing}$ CPUs ($N_{opt}$ =27, $N_{routing}$=16 over the study domain). Over the Iberian
Peninsula, the range of $x$ is defined between 0 and 20 which is determined by $Q_{fg}$ and $Q_{obs}$.

In order to check the impacts of prior information $x_{prior}$ on the optimization convergence

time, the $x_{prior}$ is set to a constant value '1' ($x_{prior\_1}$) or a 'pre-estimated-prior' ($x_{prior\_ref}$, defined as
the ratio of $Q_{obs}/Q_{fg}$), separately. The optimal $x$ values are assigned over the whole study domain.
The $x$ of the sub-catchment without GRDC station available is set to 1 (no correction). The
climatology values (e.g., over 1979-2014) are applied to fill the observation missing values over
certain period. In case of more than one GRDC stations locate in the same model grid, the averaged
correction factor is used.

The optimization results are not sensitive to the choice of $x_{prior}$, but the convergence time

indeed depends on $x_{prior}$. Fig. 3a shows that the $x_{prior\_ref}$ method requires less iteration to converge
than $x_{prior\_1}$ (7 and 15-20 iterations, respectively). The value of the cost function of $x_{prior\_ref}$ method
is lower than that of $x_{prior\_1}$ for all iteration steps. The normalized bias (*Norm_BIAS*) of discharge
after 7 iterations is less than 0.3 for the $x_{prior\_ref}$ method, while it is larger than 0.6 over most south
regions for $x_{prior\_1}$ (Figs. 3b and 3c). The oscillation of $J$ at the steps 3 and 5 could be due to the
fact that the calculation of the gradient of $J$ by finite difference is not optimal. It is also possible
because the L-BFGS-B explores partly the physical range during the first few iteration to estimate
the Hessian of the cost function for convergence.
$$Norm\_BIAS = \frac{Q_{sim} - Q_{obs}}{Q_{obs}}, \tag{8}$$

We choose $x_{prior}$ set by $x_{prior\_ref}$ for $n$ years ($n$=10, 1980-1989) experiment with iteration

number $k$ being 15 and number of correction factor $m$ (i.e., the number of GRDC station) being 27.
The $x$ values vary with different years. Due to the slow variation in aquifer levels, a spin-up is
necessary before optimization to get equilibrium of aquifer levels in LSM. The spin-up creates the
aquifer initial states ($A^0_{corr}$, $A^1_{corr}$, $A^2_{corr}$, ... , $A^{10}_{corr}$) at the start of the assimilation cycles over each
ORCHIDEE model grid (Fig. 4), making it adapt to the bias corrected aquifer states.
$$\frac{dA^i_{corr}}{dt} = \left[\int_S x(R_2 + D_2)\right] - Q_{corr,2} + Q_{corr,1}, 0 \leq i \leq 10 \tag{9}$$

To test different assumptions of errors in initial conditions, we implemented different

optimization methods with each method results in a group ($m{\times}n$) of optimal $x$ (Fig. 4). In method
1, the optimization is carried out year by year with one-year spin-up for each iteration ('Y1SP1'
here after). The $x$ of the optimization year is applied during simulation. The method 2 is similar
with Y1SP1 except that it uses optimized aquifer levels from the previous year ('Y1SP0' here
after). This method assumes the finial state variables (aquifer levels) of the optimal solution at the
current optimization year is the best initial condition for the following assimilation year. In method
3, the optimization is done over 10 years continuously with 1-year spin-up at the beginning of each
10-year simulation ('Y10C' here after). The Y10C optimizes 270 $x$ over 10 years together, while
the Y1SP1 and Y1SP0 optimize the 10 years separately with 27 $x$ each year. The 'river routing'
model running years required by the three methods are 8100 (=$m{\times}2{\times}n{\times}k$), 4050 (=$m{\times}n{\times}k$) and
44550 [=$m{\times}n{\times}(n+1){\times}k$], respectively. Take the Y1SP0 for example, in each iteration, the
correction factor $x$ is perturbed by $m$ times. For each perturbation, the ORCHIDEE river routing
model runs once with one $x$ (e.g., $x_i$ at the $i$th sub-catchment) being perturbed while the $x$ of other
sub-catchments are kept the same. Therefore, the total number of years required for $m$ stations, $n$
iterations and $k$ years assimilation is $m \times n \times k$. For all experiments, the optimization is carried out at
daily scale, and the diagnostics are performed for annual averages where we assume the water
storage variation is neglectable.
In order to further identify the impacts of atmospheric forcing on optimizations (e.g.,
optimal correction factor $x$), we measure the 'Uncertainty' of the variable ('*var*' in equation; '*var*'
refers to $x$, corrected evaporation, etc.) by Eq. (10). The higher the 'Uncertainty' is, the larger the
uncertainty is. The 0 value means that all the three '*var*' values are equal.

$$Uncertainty(var) = \frac{|var_1 - var_2| + |var_2 - var_3| + |var_1 - var_3|}{3} \qquad (10)$$

**3. Results and discussions**
**3.1. Evaluation of river discharge without assimilation**
Fig. 5 displays the first guess simulation forced with different atmospheric forcing:
WFDEI_GPCC (Figs. 5a-5b), WFDEI_CRU (Figs. 5c-5d), and CRU_NCEP (Figs. 5e-5f). The
*Norm_BIAS* and correlation coefficient (computed by the annual mean values) are used to measure
the qualities of the simulated discharge. The diagnostics at each GRDC station are spread to the
entire upstream basin which contributes to the errors in discharge at downstream. The correlation
coefficient between FG (forced by WFDEI_GPCC and WFDEI_CRU) and observation is greater
than 0.6 over most regions, but it is less than 0.2 over certain regions (e.g., middle and southeast
of the Iberian Peninsula Figs. 5a and 5c). The correlation coefficient obtained by using
CRU_NCEP forcing is less than 0.2 for most regions (middle and west of the Iberian Peninsula),
which is worse than the simulation from WFDEI_GPCC and WFDEI_CRU. Wang et al. (2016)
also show the relatively poor performance of CRU_NCEP in simulating global land surface
hydrology and heat fluxes by using the Community Land Model (CLM4.5). The spatial pattern of
the absolute bias in river discharge varies with the atmospheric forcing (not shown). The
normalized bias is then applied to measure the river discharge simulation. The *Norm_BIAS* in
discharge shows consistent spatial distribution for simulations of three forcing. The *Norm_BIAS*
(positive) is higher than a factor of 1.5 over south and northeast of the Iberian Peninsula, which
means the overestimation of river discharge. The *Norm_BIAS* is small (within +/- 0.3) over north,
west and southeast of the region (Figs. 5b, 5d and 5f).

**3.2. Comparison of the three optimization strategies forced by WFDEI_GPCC**

We apply the three assimilate approaches (Y1SP1, Y1SP0, Y10C) to ORCHIDEE
simulations to correct the bias in discharge simulation by WFDEI_GPCC forcing. Fig. 6 (left)
displays the geographical distribution of the average correction factor $x$ obtained after the
assimilation. The $x$ values range between 0 and 1.5 over the study domain. The perfect discharge
simulation corresponds to $x$ equal 1. The $x$ value lower than 1 means the discharge in FG
(WFDEI_GPCC) is overestimated and thus a decrease of $R$ and $D$ is required, and vice versa for $x$
being higher than 1. The further the $x$ away from 1, the larger the corrections of runoff and drainage
are. The three methods display similar spatial distribution pattern with $x$ being less than 0.5 over
south and east of the Iberian Peninsula and $x$ being higher than 1 over north of the Iberian Peninsula.
This spatial distribution of $x$ is highly consistent with the pattern of *Norm_BIAS* in FG (discharge
overestimated in south and northeast, underestimated in north).
Fig. 6 (central column) shows the correlation coefficient between corrected discharge and
GRDC observations. After assimilation, the correlation of the optimized discharge and
observations is larger than 0.8 over most regions. The correlation coefficient for assimilated
discharge and observation is less than 0.6 (but higher than 0.4) over some regions and seems very
dependent on the forcing. This is probably because there is a contradiction of $x$ between the
upstream and downstream stations and thus the method has difficulties finding a compromise (e.g.,
over the Ebro basin). In general, the regions with low correlation coefficient are forcing dependent,
while the regions with high correlation coefficient are very consistent among different forcing. Fig.
6 (right) gives the *Norm_BIAS* in discharge between assimilations and observations. After
assimilation, this positive bias in river discharge has been significantly reduced (within ±0.3). It
should be mentioned that the $x_{prior\_ref}$ is able to capture the general distribution pattern of optimal
$x$, but the performance of river discharge estimation is significantly improved through optimization.
The role of optimization is to find an appropriate correction factor when there are several basins
(with observations) overlaps at upstream

A common validation approach is to compare the assimilated river discharge with other independent data sources. However, the river discharge observations are limited, and the GRDC is the only comprehensive river discharge datasets at global scale so far. To overcome this limitation, the assimilated river discharges are also validated over the catchments where the GRDC stations are discarded during assimilation. Fig. 7 shows the annual mean of river discharge over the Alcala Del Rio station (-5.98ºW, 37.52ºN) on the Guadalquivir river (locates at southwest of Spain) before and after correction. The observation of this station is not assimilated due to its large upstream area difference (15.53%>10%) between model (55635 $km^2$) and GRDC (46995 $km^2$). The overestimated discharge simulated by the model at this station is also corrected because it benefits from the correction factor estimated at the Cantillana station (-5.83ºW, 37.59ºN; 44871 $km^2$) which locates at the 15.3 km upstream of Alcala Del Rio station of the Guadalquivir River (southwest of the Iberian Peninsula). Between the two stations, there are several tributaries flow to Alcala Del Rio station, which leads to different annual mean river discharges at Cantillana (49.7 $m^3/y$) and Alcala Del Rio stations (94.8 $m^3/y$). This result illustrates that this approach is able to correct the river discharge over the entire basin. The discharges for certain sub-basins without assimilated observations (e.g., observation unavailable or GRDC stations discarded) are corrected by $x$ as well. Although the validation datasets are from the same GRDC source, they are from other independent observation stations thus can be seen as an independent validation ('first order validation').

In summary, all the three methods (Y1SP1, Y1SP0, and Y10C) are able to improve the river discharge simulation by ORCHIDEE LSM. The correlation coefficient and *Norm_BIAS* in discharge obtained from the three methods are generally consistent. The correlation coefficient of Y10C method in northeast is lower than that of Y1SP0 and Y1SP0, which is probably resulted from its poor quality of atmospheric forcing. The Y1SP0 consumes less computing time than Y1SP1 and Y10C, and it does not worsen the optimization results. By compromising between the accuracy of results and the computing time, we choose Y1SP0 method for the further assimilation.

The above assimilations are performed with the same forcing (WFDEI-GPCC) by assuming the errors in discharge are caused by model defect (e.g., model parameterization, model structure, etc.). The uncertainties of simulated discharge also result from the atmospheric forcing. The role of atmospheric forcing in assimilation is discussed in following section.

**3.3. The sensitivity of the optimizations to atmospheric forcing**

In order to understand the response of the optimizations to different atmospheric forcing with different precipitation sources, the ORCHIDAS was also run with WFDEI_CRU and CRU_NCEP forcing using Y1SP0 optimization strategy. Using two other different forcing for the assimilation can allows us to understand how important the forcing uncertainty affects the correction factor. The multi-year mean correction factor $x$ obtained from WFDEI_CRU (Fig. 8a) CRU_GPCC (Fig. 8b), and WFDEI_GPCC (Fig. 8c) displays quite consistent spatial patterns. The coverage of low correction factor (blue in Figs. 8a-8b, corresponds to large correction) obtained from CRU-NCEP is larger than that obtained from WFDEI_CRU and WFDEI_GPCC. This is because the positive bias in discharge of FG simulation forced by CRU-NCEP is larger than that by WFDEI_CRU and WFDEI_GPCC. Besides the atmospheric forcing, the uncertainties could also origin from boundary condition (e.g., topographic or other land surface features), model parameter, model structure or missing processes. For all forcing, the $x$ is less than 0.3 (but greater than 0) over south, which implies that the error in discharge is probably resulted from the missing model processes (human activity). Over north, the $x$ are close to 1 (discharge well simulated) for all the three forcing, which indicates the correction comes from model 'random' error (nature discharge) rather than the system error (e.g., missing processes).

The uncertainty of $x$ by three forcing is small for most regions (Fig. 8d). The high uncertainty of $x$ over the Adoure (southwestern France) and the Chelif (in Algeria) river basins corresponds to the large uncertainty in the different atmospheric forcing. This result demonstrates the obtained correction factor $x$ is robust in spite of using different atmospheric forcing. This is also demonstrated by comparing the precipitations between the three forcing and the IB02 dataset. Compared to the IB02, all the three forcing overestimate rainfall in the Iberian Peninsula (Figs. S1a-S1c), but none of these error patterns resembles that of the proposed $E$ correction (Figs. 9e-9g). Unlike the pattern of the correction factor (Figs. 8a-8c), the ratios of annual mean precipitation between the three forcing and the IB02 are higher than 1 over most regions (Figs. S1d-S1f). Therefore, the precipitation forcing error is not likely the dominant factor in determining the correction factor distribution.

In summary, the assimilation approach is able to correct errors in lateral water balance
despite using different forcing. Recalling that the corrected $R+D$ (through $x$) and the precipitation
are known, we then transfer the optimal correction factor $x$ to the vertical water balance equation
(Eq. 5) to derive the bias corrected evaporation. This will enable us to understand the impacts of
assimilation on evaporation.
**3.4. Evaporation estimations through the optimal correction factor**

The evaporation of FG simulation by different forcing show quite consistent spatial
distribution (Figs. 9a-9c) and small uncertainty (<0.2 mm/d, Fig. 9d) with the value being higher
over north than south. The change of evaporation ($dE$) induced by the correction is consistent for
three forcing (Figs. 9e-9g) with low uncertainties (Fig. 9h). It should be mentioned that the
evaporation for the regions without GRDC stations are not corrected (i.e., correction factor $x$ equals
1) such as southern France, western Portugal, and northwest, south and southeast of Spain (blank
regions in Fig. 8). The $dE$ is positive (around 0.2 to 0.4 mm/d) over south and northeast where the
evaporation is underestimated in FG. Cazcarro et al. (2015) show large blue water footprint
(volume of surface and groundwater consumed for production an item) of human activity over
south (Jaén, Sevilla, and Malaga provinces), northeast (Palencia, Burgos, La Rioja, Navarra and
Valladolid provinces), north (Tarragona province) and middle (Toledo province) of Spain (Map.
1 of that paper). The large $dE$ over south and northeast obtained in current study is consistent with
the blue water footprint of Cazcarro et al. (2015). Figs 9i-9k plot the change of the ratio of water
demand ($dE$) and water supply ($R+D$). This ratio measures the degree of water shortage. The
greater the ratio, the higher level of water shortage. The ratio is larger over south and northeast of
Spain, which is consistent with the results from other studies that measures the water deficits
(Rodríguez-Díaz et al., 2007) and water exploitation index (Pedro-Monzonís et al., 2015) in Spain.
Since we assume that the missing human processes is the main error in ORCHIDEE, the $dE$ and
$dE/(R+D)$ indicate the changes induced by human processes. The spatial patterns of $dE$ and
$dE/(R+D)$ are quite consistent with human water exploitation, thus the model missing processes
(e.g., human water usage) is considered as the dominant contribution to $x$.

We also tested the possibility of improving the river discharge estimation by using a annual
constant correction factor to evaporation ($X_{Ecorr}$), which can be derived from Eq. (6).

$$X_{Ecorr} \approx \frac{E + (1 - x) \cdot (R + D)}{E},$$ (11)

$$E_{corr} = X_{Ecorr} \cdot E$$ (12)

Although the Eqs. 11-12 are able to improve river discharge estimation by modifying soil moisture, the energy and water balance are not conserved. One solution could be to run the full ORCHIDEE LSM in the assimilation system with the same cost function as Eq. (7). In this way, the intermediate variables are adjusted towards optimal river discharge with the modification of evaporation. This approach executes the full ORCHIDEE model thus is very time consuming and is beyond the scope of the current study.

**3.5. The inter-annual variation of correction factor and water cycle**

**3.5.1. The inter-annual cycles**

All the results so far are obtained by averaging multi-year mean values which provides us the bias correction information at spatial scale. To understand the inter-annual cycles of the correction and its possible contribution, we analyze the assimilation results over two stations at south of Spain where the discharge correction is large during the period of 1980 - 1989 (Fig. 8).

The Puente De Palmas station locates on the Guadiana River (southwest of the Iberian Peninsula) with an upstream area of 48515 km². The three FG simulations (with different forcing) significantly overestimate the river discharge and the runoff coefficient (ratio of discharge and precipitation), while the FG(WFDEIG) and FG(WFDEIC) underestimate the inter-annual variability comparing with observations (Fig. 10a-10b). The standard-deviation of the annual means for observation, FG(WFDEIG), FG(WFDEIC) and FG(CRUN) are 33.8 m³/s, 28.8 m³/s, 25.2 m³/s and 34.3 m³/s, respectively. One reason could be the variation of water usage by irrigated agriculture which occupies 90% of the blue water usage (surface water and groundwater) in this semiarid basin (Aldaya and Llamas, 2008) or model errors. Besides, there are many interconnected wetlands and structurally complex hydrogeological boundaries between the two upper-Guadiana aquifer in the upper Guadiana River basin (Van Loon and Van Lanen, 2013). These complex features are difficult to represent in model thus large bias exist in river discharge

of ORCHIDEE. The correction factor corrects these model defects (Fig. 10c) and it demonstrates
good skill in correcting the inter-annual variability of discharge and runoff coefficient (Fig. 10a-
10b).
The Masia De Pompo station (17876 km$^2$) is on the Jucar River (southeast of Spain). The
observations over the year 1983, 1988-1989 are obtained from the climatology values due to
the unavailability of GRDC data during this period. During 1980-1989, the inter-annual
variation of observed discharge (and runoff coefficient) and FG simulation is quite inconsistent
(Figs. 10d-10e). This is probably caused by the surface water usage which occupies about 55%
over this basin (Kahil et al., 2016). Most of them are used for agriculture (>80%) and urban
(>10%). Although the improvements in assimilated discharge are small, the correction factor is
able to capture the inter-annual variability in observations (Figs. 10d and 10f).
In summary, the inter-annual variation river discharge of FG simulation and
observations does not agree each other over the Guadiana River basin and the Jucar River basin
during 1980-1989. The human water usage (e.g., groundwater or surface water extraction)
process, which is neglected in current ORCHIDEE model, is likely to play an important role in
river discharge variation. The optimized correction factor (varies each year) improves the inter-
annual variability of the modelled river discharge.
**3.5.2. The geographical distribution**
To further understand the inter-annual variability of corrections over the entire Iberian
Peninsula region, Fig. 11 plots the spatial distribution of inter-annual variability of correction
factor *x* and river discharge which is quantified by coefficient of variation as used by Déry et al.
(2011) and Siam and Eltahir Elfatih (2017). In FG (WFDEI_GPCC) simulation, the inter-annual
variation of discharge is lower than 0.4 over most regions, which indicates an underestimation of
inter-annual variability of river discharge in FG. The inter-annul variability of discharge is
increased after assimilation over south and northeast. This change could be attributed to the
fluctuation of correction factor (human water usage) over these regions. This result agrees with the
results (Map. 6) of Cazcarro et al. (2015) with more large dams in south and northeast (nature
discharge greatly affected by human) than northwest of Spain (nature discharge less affected by
human). The inter-annual variability of correction factor $x$ and discharge for Y1SP0 (CRUN) is
different from others, which mainly results from the different atmospheric forcing.

### 3.6. Comparison of bias corrected evaporation with GLEAM data

In order to evaluate the bias corrected evaporation, Figs. 12a-12h compare the GLEAM
product (v3.1a) with FG and with bias corrected $E$ by assimilation using WFDEI_GPCC,
WFDEI_CRU and CRU_NCEP forcing. Due to the unavailability of parts of GLEAM's
atmospheric forcing (e.g., air pressure, air humidity, air speed, etc.) and difficulty of maintaining
a coherence with other forcing, the assimilation system does not run with GLEAM's precipitation
input. We find large difference between GLEAM and FG, which indicates that the evaporation is
quite uncertain for different estimations. The geographical distribution and magnitude of
difference in $E$ between GLEAM and FG is highly consistent with that between GLEAM and bias
corrected values by using different forcing (Figs. 12a-12c, and 12e-12g). The systematic negative
difference is higher than the uncertainties of bias corrected $E$ with different forcing (Figs. 12d and
12h). Parts of the differences are explained by the lower $P$ of GLEAM than ORCHIDEE forcing
(Figs. 12i-12l). Generally, the $P$-$E$ (in mm/d) of GLEAM is higher than bias corrected value
associated with small uncertainties (Figs. 12m-12t). Because the bias corrected $P$-$E$ are corrected
by GRDC observed river discharge, the $P$-$E$ (≈river discharge) of GLEAM is very likely to be
higher than GRDC observations over the Iberia. This result indicates that some processes are
probably also missing in GLEAM v3.1. We also compared our bias corrected $E$ with GLEAM v1
data (Miralles et al., 2011), and we find the $P$-$E$ between GLEAM v1 and bias corrected values
are quite consistent for different forcing. The results are quite consistent when comparing the
corrected $E$ with several other products which are obtained by using different methodology and
forcing (e.g., Jung et al., 2009; Vinukollu et al., 2011; Mueller et al., 2013). Considering the
availability of $P$-$E$ for GLEAM data which allows to compare it with the bias corrected value, only
the results of GLEAM are shown.

### 4. Conclusions

There has been several studies working on estimation of fresh water input from continent
to ocean (e.g., the Mediterranean Sea) based on observation or modelling approach (e.g., Boukthir
and Barnier, 2000; Mariotti et al., 2002; Struglia et al., 2004; Peucker-Ehrenbrink, 2009; Ludwig
et al., 2009; Szczypta et al., 2012). However, these estimations are limited either by the coarse
temporal resolution for observation approach or by the non-comprehensive representation of
physical processes (e.g., human activities) for modelling approach. As a result, the fresh water
estimations are accompanied with large uncertainties among varies studies. This proposed
methodology aims to improve the estimation of continental water cycles by merging the merits of
observations and modelling approach through data assimilation.
The basis of the method is the vertical and lateral water balance equations. The method
assumes that the precipitation minus evaporation from the model simulation is an appropriate first
guess so that all the errors in river discharge end up with runoff and drainage. Under this
assumption, the river discharges simulation at river outlet are expected to be improved by
correcting the runoff and drainage (inputs for river routing model).
The idea is achieved by embedding a river routing scheme of ORCHIDEE LSM and GRDC
river discharge observations into a data assimilation system (ORCHIDAS). The system can run
with multi-level parallel computing mode (both the routing model and the optimization are
parallelized). The river discharge is optimized through applying a correction factor $x$ to model
runoff and drainage which translates errors in estimated $P$-$E$.
The method has been explained through its application over the Iberian Peninsula with 27
GRDC stations during 1979-1989 with $x$ values being different each year. Main conclusions are:
First, the optimization results are not sensitive to $x$ prior information $x_{prior}$, and assimilation
strategies, but the setting of $x_{prior}$ by a 'pre-estimated-prior' (defined as $Q_{obs}/Q_{fg}$) indeed converges
faster than other $x_{prior}$ values. The method Y1SP0 (the model spin-up uses the optimal aquifer
levels of previous optimization year) demonstrates high computing efficiency and comparable
discharge accuracy comparing with the other two methods (Y1SP0, Y10C), thus the Y1SP0 is
recommended (e.g., over the full Mediterranean catchment). Second, the largest correction of
discharge is found over south and northeast of the Iberian Peninsula. These regions are
characterized by large blue water footprint with large groundwater and surface water usage by
human activity. It implies that most of the corrections by $x$ represents the missing human processes
(at least in the south of study domain). This is consistent with the fact that ORCHIDEE model
neglects the human processes (e.g., dam operation, irrigation, etc.). The discharge correction over
north of the Iberian Peninsula is relatively small, where is mainly due to model systematic error.
The correction factor $x$ can also cover errors in the model structure, model parameter, or boundary
conditions (e.g., land surface characteristics imposed to the model). Third, the assimilated
discharges reveal lower bias (from >100% to <30%) and higher inter-annual variability (due to the
fluctuation of water usage) than uncorrected ones. Fourth, the bias corrected evaporation are
compared with the GLEAM v3.1a product. The $E$ of GLEAM is lower than the optimized $E$, while
the $P$-$E$ of GLEAM is higher than the optimized values. This different $P$-$E$ could be caused by the
different $P$ forcing and the missing processes in the GLEAM model.
The method takes into account both gauged rivers (usually large rivers) and un-gauged
rivers, and it provides discharge estimates at daily scale from 1980 to 2014 with the time range
depend on atmospheric forcing. By using the correction factor of adjacent catchment, this method
also improves the river discharge simulation for the catchment without assimilating observations.
Besides, this method fills the gap of the data missing period (e.g., war, instruments, etc.) by
climatology values, thus the data are complete over the whole period. The proposed method is
supposed to be superior to the simple water-balance methods, because a LSM estimates $E$ at sub-
diurnal scales with physically based equations and takes advantage of spatial distribution of the $P$
and $P$-$E$.
The result implies the necessity of parameterizing the human water uptake process in the
ORCHIDEE LSM. Besides, the poor quality of the river discharge observations (e.g., 68% stations
are discarded over the Iberian Peninsula) calls for a high quality data. The optimized correction
factors $x$ are model and atmospheric forcing dependent. It is encouraged to apply this assimilation
method to other models, which will allow us to identify the sources of errors (e.g., model missing
process or forcing data). To improve the calculation efficiency, this study uses annual mean
correction factors without considering its seasonal variation thus the seasonal discharges do not
improved. One issue of the $x$ optimization could be the equifinality with a number of optimized $x$
result in the similar river discharge at downstream. Future developments can be made towards
generating ensemble optimal $x$ to better assess the uncertainties associated to each parameter $x$.
This assimilation method can be applied for water cycles studies, data inter-comparison, and
riverine fresh water estimation over other basins (e.g., the full catchment of the Mediterranean sea).

## Acknowledgments


The authors gratefully acknowledge financial support provided by the STSE WACMOS-
MED (Water Cycle Multi-mission Observation Strategy for the Mediterranean) project under ESA
(Grant No. 4000114770/15/I-SBo) and the Earth2Observe (Global Earth Observation for
Integrated Water Resource Assessment) project of the FP7 (Grant No. 603608). The ClimServ
computational facilities at IPSL were used to perform all the simulations. The authors also thank
the valuable and constructive comments from Emanuel Dutra (Lisbon University) and another
anonymous reviewer.

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

 **Figure captions:**

**Figure 1.** (a) The illustration of correcting river discharge ($Q$) simulation (simulation in blue solid
dot, observation in red star) by applying correction factors ($x$) to runoff and drainage over different
basins. The basin 1 and basin 2 are represented in yellow and blue, respectively. (b) The model
framework of the river discharge assimilation. The blue and red parts are run for 'First Guess' and
for assimilation, respectively.
**Figure 2.** The river network (blue lines) and the GRDC stations (solid dots represent the 27
qualified stations and the gray triangles represent unqualified stations) over the study domain.
**Figure 3.** (a) The variation of cost function $J$ (unit: 1; logarithmic y-axis) with iterations for $x_{prior\_1}$
('$x_{prior} = 1$', in blue) and for $x_{prior\_ref}$ ('$x_{prior}$ = pre-estimated-prior', in red). The iterations 6-15 are
enlarged in the window (normal y-axis). The *Norm_BIAS* of optimized river discharge after 7
iterations for $x_{prior\_1}$ (b) and for $x_{prior\_ref}$ (c).
**Figure 4.** The set-up of assimilation experiments for $n$ years ($n$=10, 1980-1989) and $k$ iterations
($k$=10) with $m$ ($m$=27) correction factors ($x$) each year ($x$ is different over years). (a) The $i$th year
($Y_i$) optimization is initialized by the end of $Y_{i-1}$ optimization; (b) the initial condition of $Y_i$
optimization is got by running $Y_{i-1}$ optimization fed with the same $x$ as $Y_i$; (c) optimizing $n$ years
together with one year spin-up at the beginning of $n$-year. The Y1SP0 and Y1SP1 divide the $n$-
year optimization into $n$ 1-year optimization periods. The blue and red colors mean optimization
and spin-up simulations, respectively.
**Figure 5.** The river discharge simulations from 1980 to 1989 using WFDEI_GPCC (1[st] row),
WFDEI_CRU (2[nd] row) and CRU_NCEP (3[rd] row) forcing. Left: the correlation coefficient of
river discharge between observations and simulations; Right: the *Norm_BIAS* of simulated river
discharge.
**Figure 6.** The optimization results from 1980 to 1989 using the three methods (1[st] row: Y1SP1;
2[nd] row: Y1SP0; 3[rd] row: Y10C) forced by WFDEI_GPCC. Left: the optimized correction factor
$x$; Middle: the correlation coefficient of river discharge between observations and optimizations;
Right: the *Norm_BIAS* of optimized river discharge.
**Figure 7.** The annual cycles of river discharge for 'First Guess' (FG) forced by WFDEI-GPCC
(black), Y1SP1 (blue), Y1SP0 (green), Y10C (yellow) and GRDC observations (red) over the
Alcala Del Rio station (-5.98ºW, 37.52ºN) on the Guadalquivir river. The dotted lines mean the
trend.
**Figure 8.** The correction factor $x$ obtained from Y1SP0 forced by (a) WFDEI_CRU, (b)
CRU_NCEP, (c) WFDEI_GPCC, and (d) the 'Uncertainty' (defined by Eq. 10) of $x$ by different
forcing. All values are averaged over 1980-1989.
**Figure 9.** The evaporation ($E$, in mm/d) before assimilation (1st line), change of evaporation ($dE$,
in mm/d) after and before assimilation (2nd line), and the ratio of $dE$ and runoff + drainage (3rd line)
for forcing WFDEI-GPCC (1st column), WFDEI-CRU (2nd column), CRU-NCEP (3rd column),
and the 'Uncertainty' (defined by Eq. 10) in different forcing (4th column) averaged from 1980 to

895  1989.

**Figure 10.** The optimization results by different atmospheric forcing (WFDEI-GPCC in black,
WFDEI-CRU in green, and CRU-NCEP in blue) over the Puente De Palmas station on Guadiana
River (a-d, -6.97ºW, 38.88ºN; 48515 km$^2$) and over the Masia De Pompo station on the Jucar river
(e-h, -0.65ºW, 39.15ºN; 17876 km$^2$): (a, d) annual river discharges; (b, e) runoff coefficient; (e, f)
optimized correction factor $x$ for the simulated/assimilated river discharge (First Guess in dark
color, Y1SP0 in light color) with respect to GRDC observations (in red) from 1980 to 1989.
**Figure 11.** The inter-annual variation of correction factor $x$ ($\frac{\sigma(x)}{\bar{x}}$; a, d, g), simulated river discharge
without assimilation ($\frac{\sigma(Q_{sim})}{Q_{sim}}$; b, e, h) and optimized river discharge ($\frac{\sigma(Q_{opt})}{Q_{opt}}$; c, f, i) for
Y1SP0_WFDEIGPCC (1st row), Y1SP0_WFDEICRU (2nd row) and Y1SP0_CRUNCEP (3rd row)
averaged over 1980-1989.
**Figure 12.** Comparison of evaporation ($E$, in mm/d, 1st line) between GLEAM (v3.1) and FG (First
Guess), as well as $E$ (2nd line), precipitation ($P$, in mm/d, 3rd line), $P$-$E$ (in mm/d, 4th line) and $P$-$E$
(relative value between 0-1, 5th line) between GLEAM (v3.1) and assimilated values using
different forcing (1st column: WFDEI-GPCC; 2nd column: WFDEI-CRU; 3rd column: CRU-NCEP;
4th column: 'Uncertainty' (defined by Eq. 10) of using different forcing) averaged from 1980 to

911  1989.

**Table 1**. The assimilation and simulation experiments

| Name | Atmospheric Forcing | Method |
| --- | --- | --- |
| FG(WFDEIG) | WFDEI_GPCC | No assimilation |
| FG(WFDEIC) | WFDEI_CRU | No assimilation |
| FG(CRUN) | CRU_NCEP | No assimilation |
| Y1SP0(WFDEIG) | WFDEI_GPCC | Y1SP0 assimilation |
| Y1SP1(WFDEIG) | WFDEI_GPCC | Y1SP1 assimilation |
| Y10C(WFDEIG) | WFDEI_GPCC | Y10C assimilation |
| Y1SP0(WFDEIC) | WFDEI_CRU | Y1SP0 assimilation |
| Y1SP0(CRUN) | CRU_NCEP | Y1SP0 assimilation |

Note: All runs are from 1980 to 1989 with 0.5º spatial resolution; FG stands for 'First Guess'.

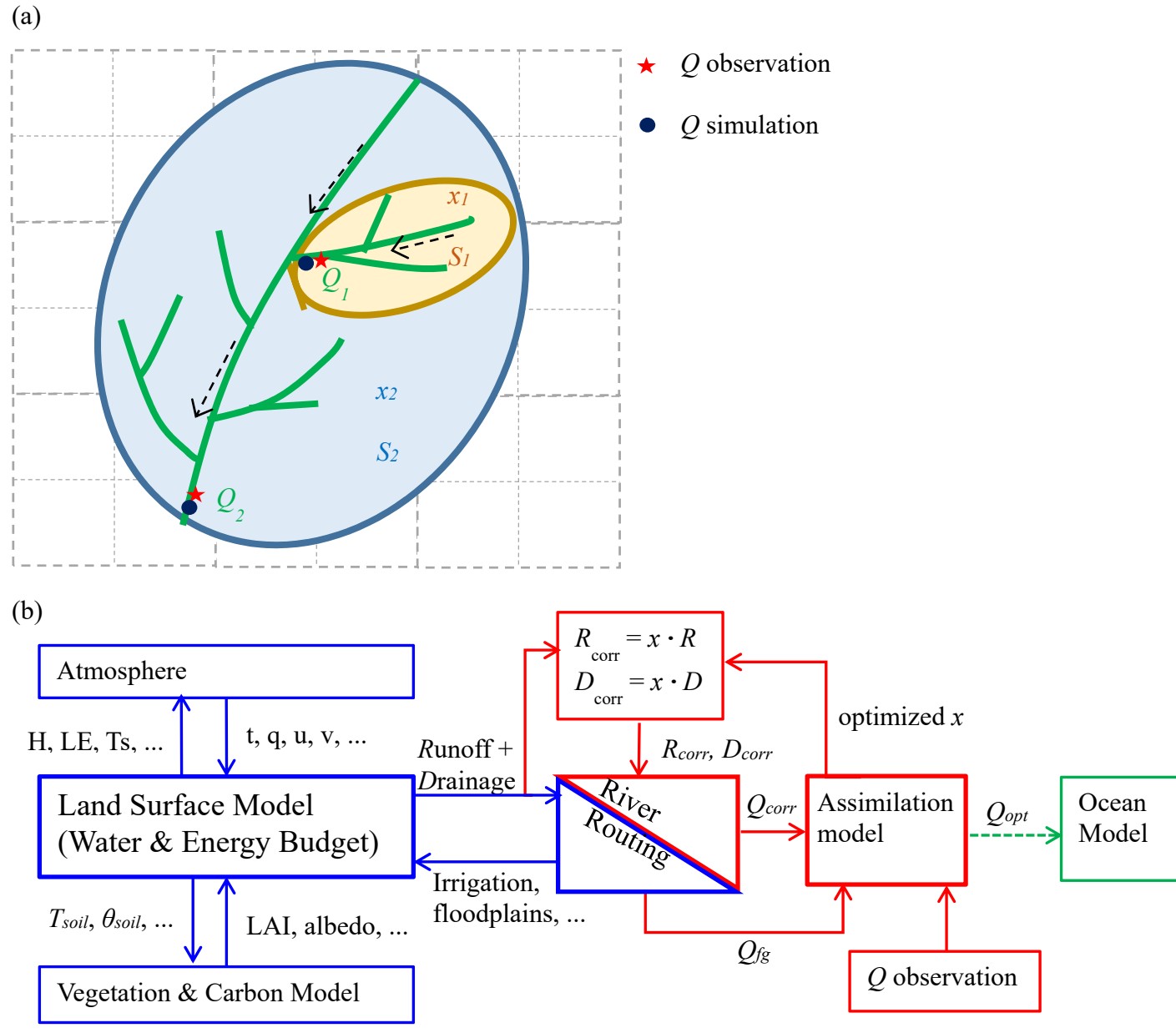

**Figure 1.** (a) The illustration of correcting river discharge ($Q$) simulation (simulation in blue solid dot, observation in red star) by applying correction factors ($x$) to runoff and drainage over different basins. The basin 1 and basin 2 are represented in yellow and blue, respectively. (b) The model framework of the river discharge assimilation. The blue and red parts are run for 'First Guess' and for assimilation, respectively.

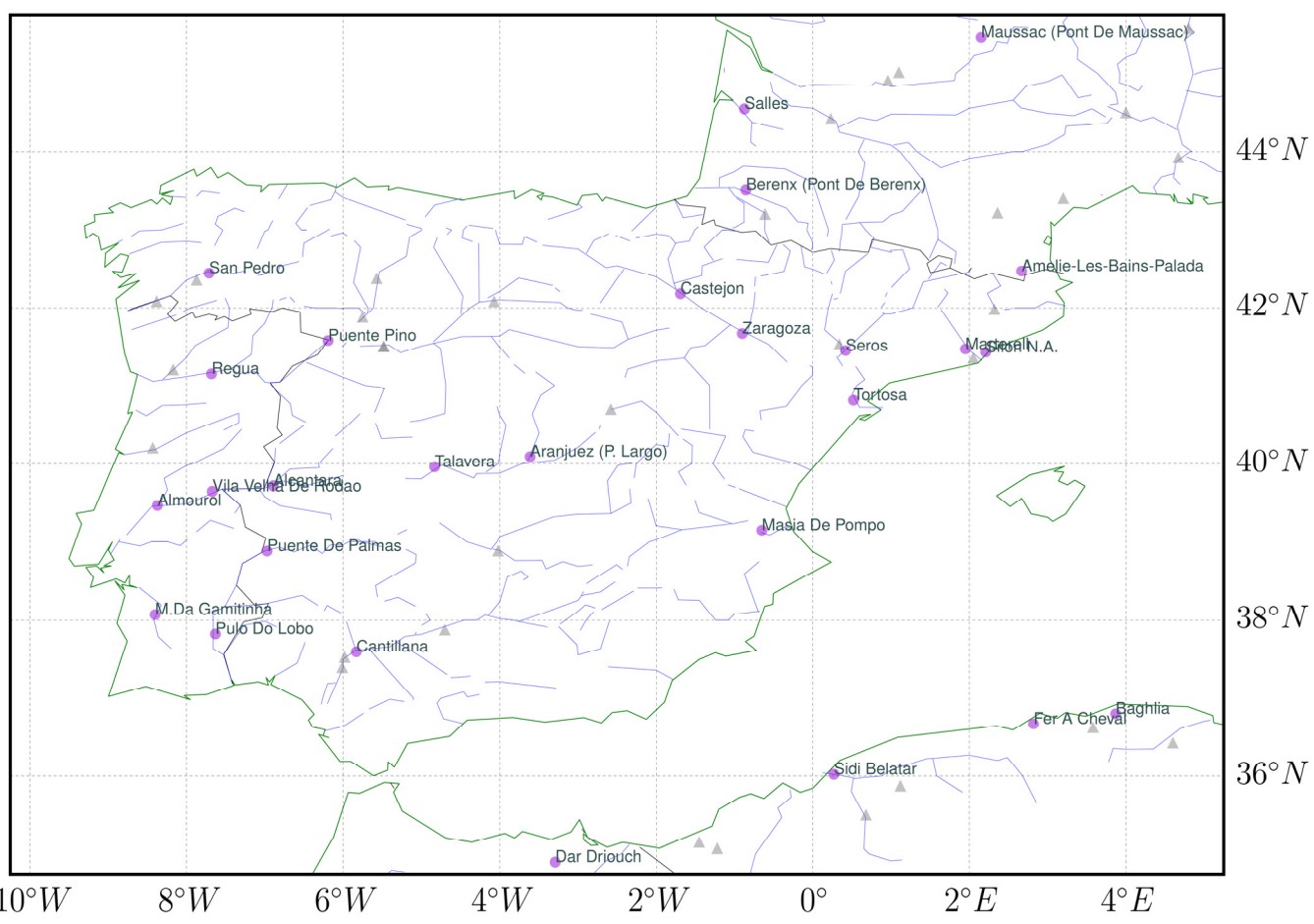

**Figure 2.** The river network (blue lines) and the GRDC stations (solid dots represent the 27 qualified stations and the gray triangles represent unqualified stations) over the study domain.

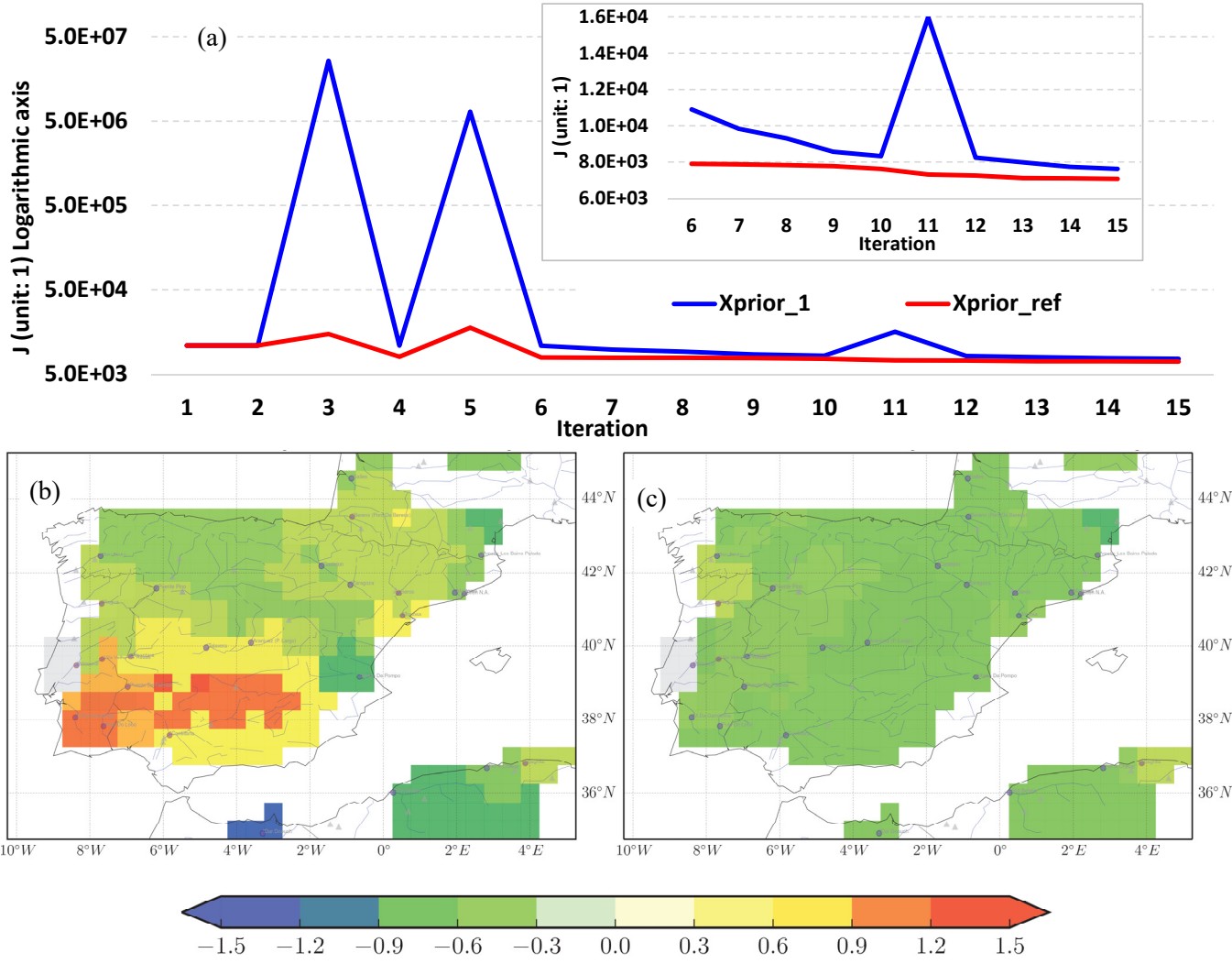

**Figure 3.** (a) The variation of cost function $J$ (unit: 1; logarithmic y-axis) with iterations for $x_{prior\_1}$ ('$x_{prior}$ = 1', in blue) and for $x_{prior\_ref}$ ('$x_{prior}$ = pre-estimated-prior', in red). The iterations 6-15 are enlarged in the window (normal y-axis). The *Norm_BIAS* of optimized river discharge after 7 iterations for $x_{prior\_1}$ (b) and for $x_{prior\_ref}$ (c).

(a) Y1SP1

$m$ factors to optimize, $k$ iterations

$A^0$ $Y_0$ $A^1$ $Y_1$    $Y_1$    $Y_2$    ...    $Y_{i-1}$    $Y_i$    ...    $Y_{n-1}$    $Y_n$

(b) Y1SP0

$m$ factors to optimize, $k$ iterations

$A^0$ $Y_0$    $A^1$ $Y_1$    $Y_2$    ...    $Y_i$    ...    $Y_{n-1}$    $Y_n$

(c) Y10C

$m \times n$ factors to optimize, $k$ iterations

$A^0$ $Y_0$ $A^1$ $Y_1$    $Y_2$    $Y_3$    $Y_{i-1}$    $Y_i$    $Y_{n-1}$    $Y_n$

**Figure 4.** The set-up of assimilation experiments for $n$ years ($n$=10, 1980-1989) and $k$ iterations ($k$=10) with $m$ ($m$=27) correction factors ($x$) each year ($x$ is different over years). (a) The $i$th year ($Y_i$) optimization is initialized by the end of $Y_{i-1}$ optimization; (b) the initial condition of $Y_i$ optimization is got by running $Y_{i-1}$ optimization fed with the same $x$ as $Y_i$; (c) optimizing $n$ years together with one year spin-up at the beginning of $n$-year. The Y1SP0 and Y1SP1 divide the $n$-year optimization into $n$ 1-year optimization periods. The blue and red colors mean optimization and spin-up simulations, respectively.

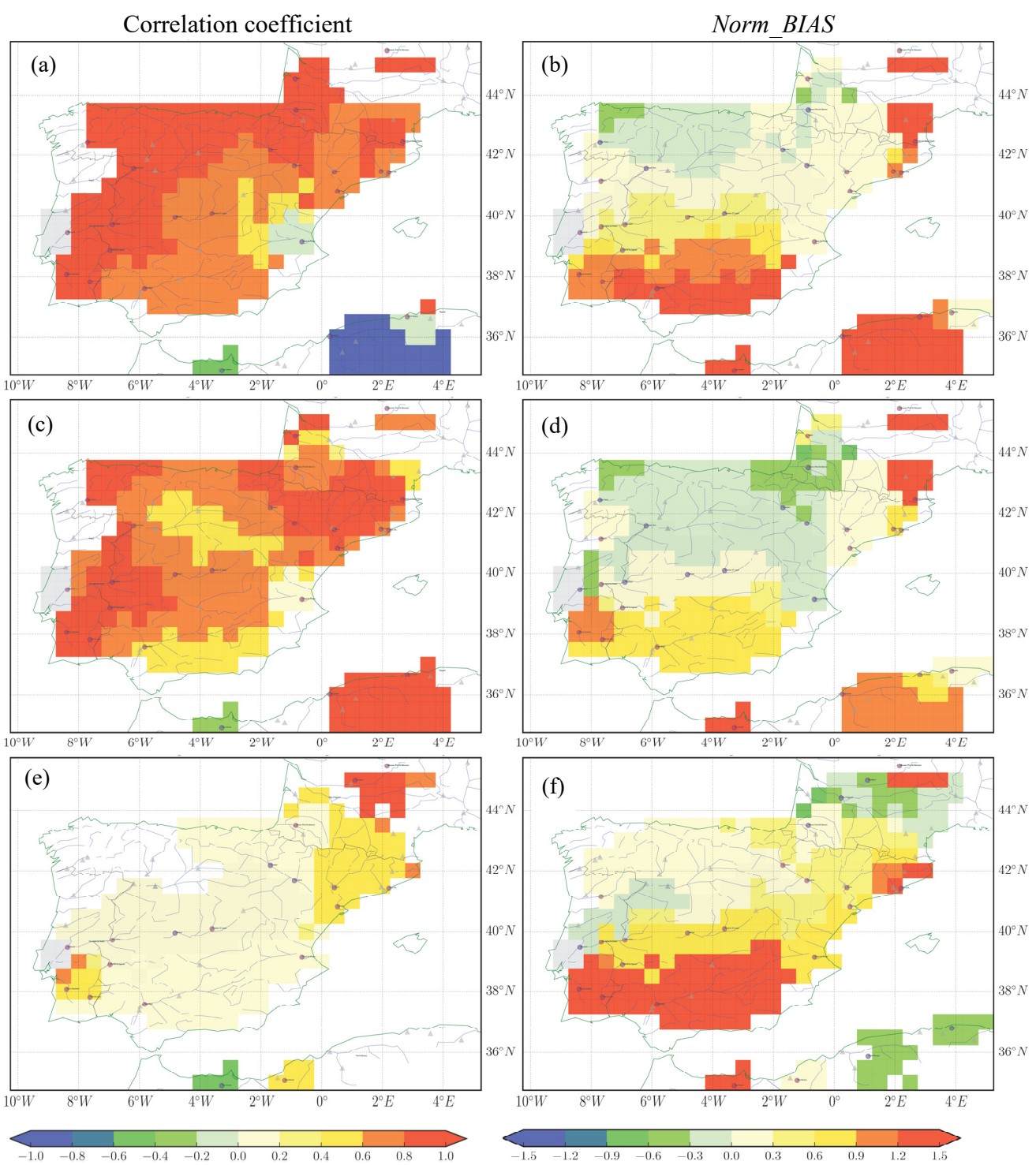

**Figure 5.** The river discharge simulations from 1980 to 1989 using WFDEI_GPCC (1st row), WFDEI_CRU (2nd row) and CRU_NCEP (3rd row) forcing. Left: the correlation coefficient of river discharge between observations and simulations; Right: the *Norm_BIAS* of simulated river discharge.

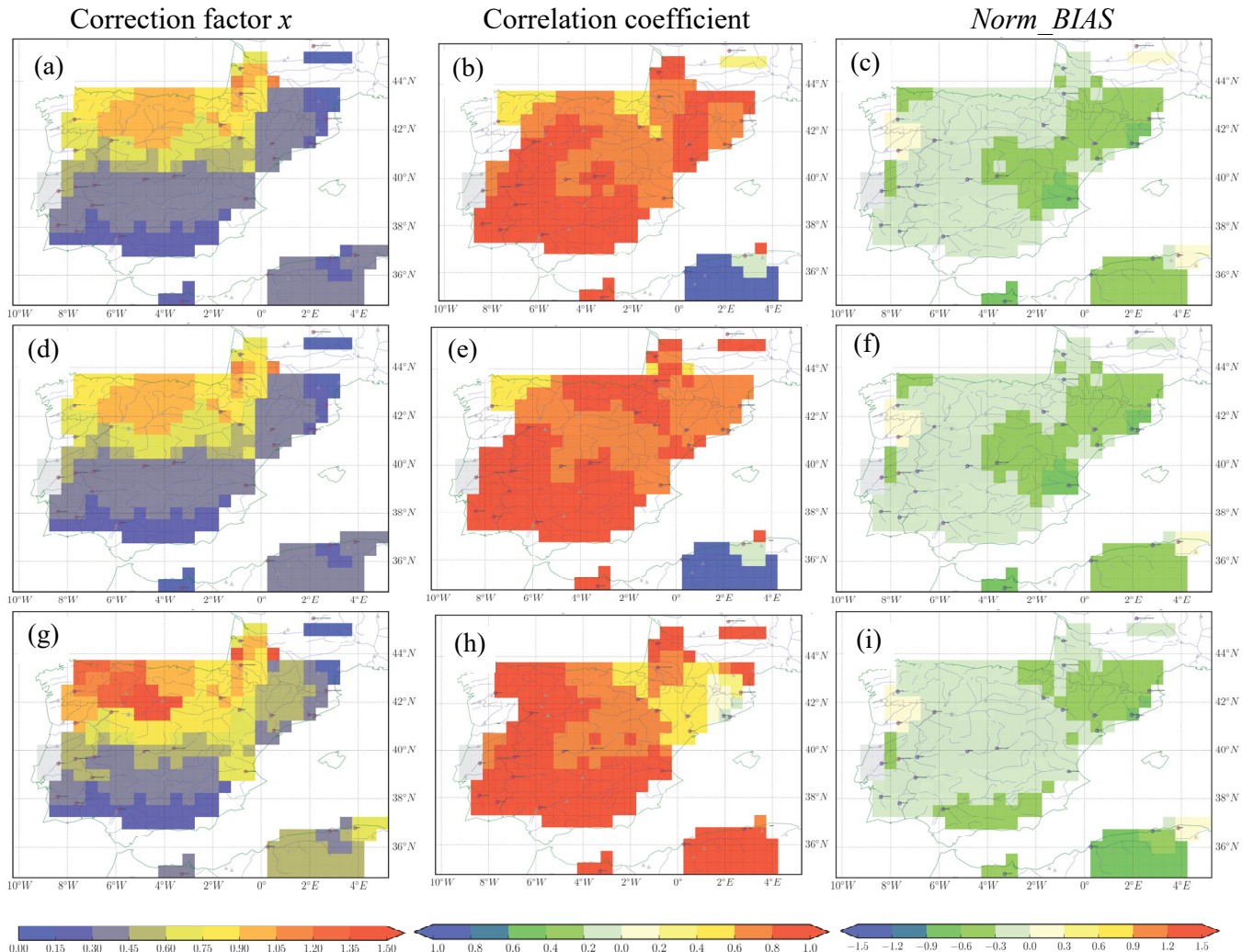

**Figure 6.** The optimization results from 1980 to 1989 using the three methods (1st row: Y1SP1; 2nd row: Y1SP0; 3rd row: Y10C) forced by WFDEI_GPCC. Left: the optimized correction factor *x*; Middle: the correlation coefficient of river discharge between observations and optimizations; Right: the *Norm_BIAS* of optimized river discharge.

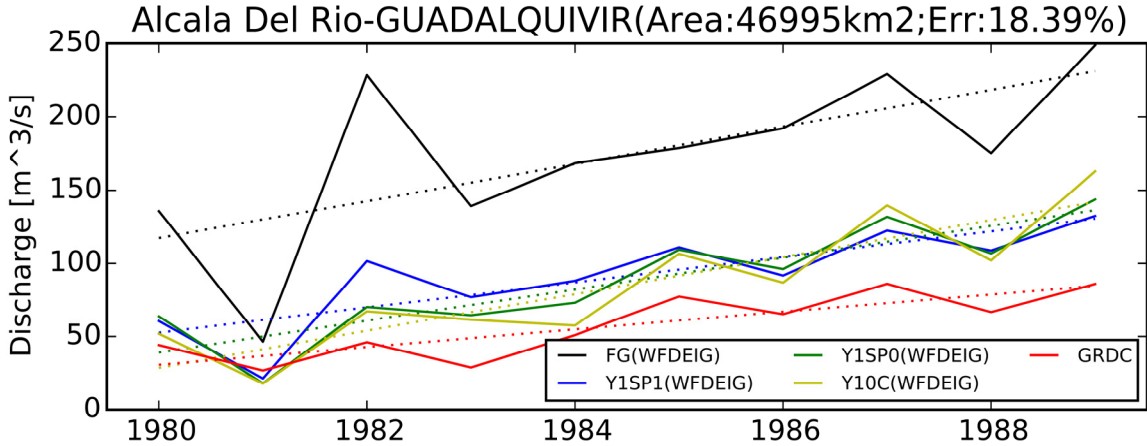

**Figure 7.** The annual cycles of river discharge for 'First Guess' (FG) forced by WFDEI-GPCC (black), Y1SP1 (blue), Y1SP0 (green), Y10C (yellow) and GRDC observations (red) over the Alcala Del Rio station (-5.98ºW, 37.52ºN) on the Guadalquivir river. The dotted lines mean the trend.

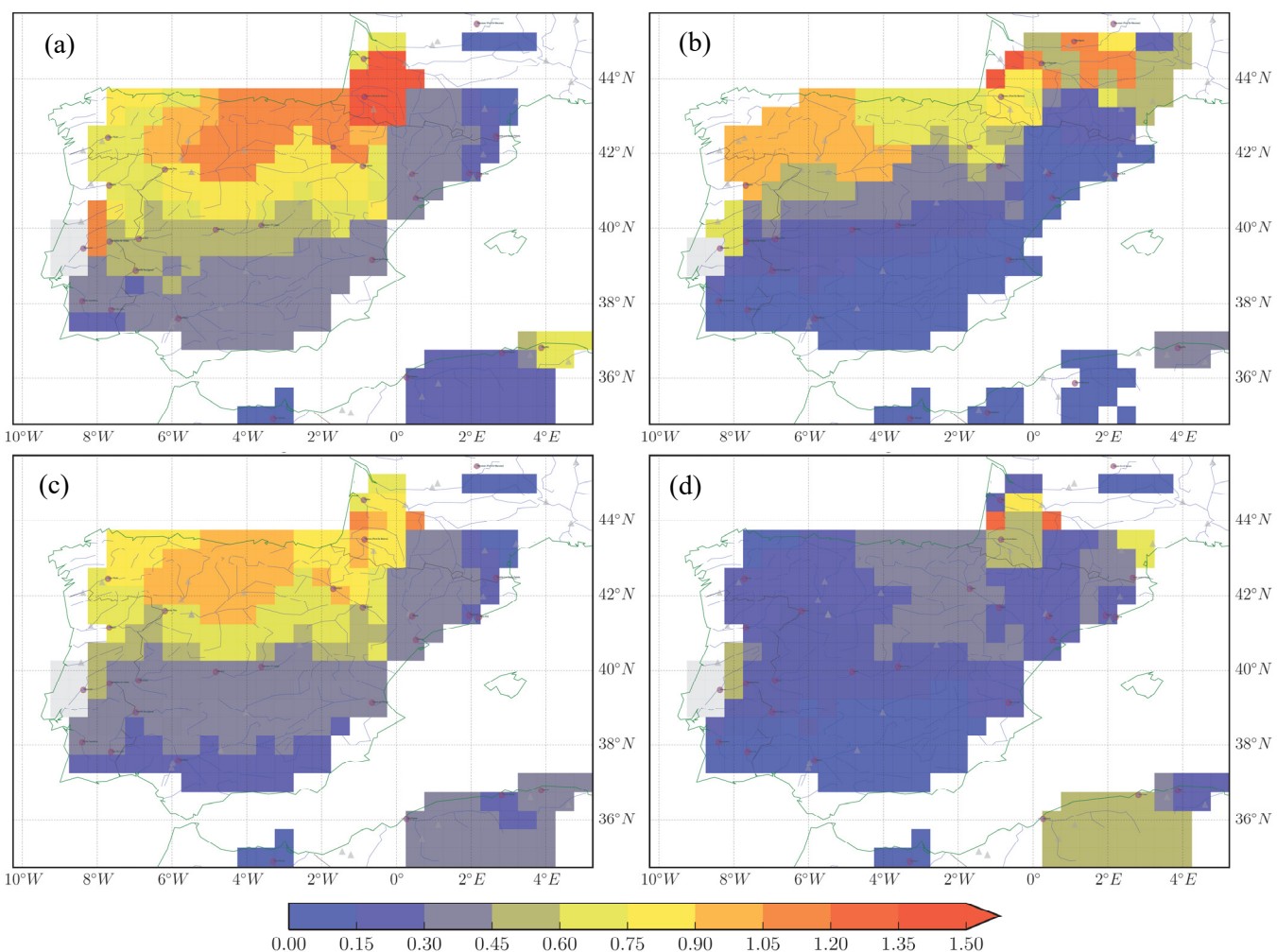

**Figure 8.** The correction factor $x$ obtained from Y1SP0 forced by (a) WFDEI_CRU, (b) CRU_NCEP, (c) WFDEI_GPCC, and (d) the 'Uncertainty' (defined by Eq. 10) of $x$ by different forcing. All values are averaged over 1980-1989.

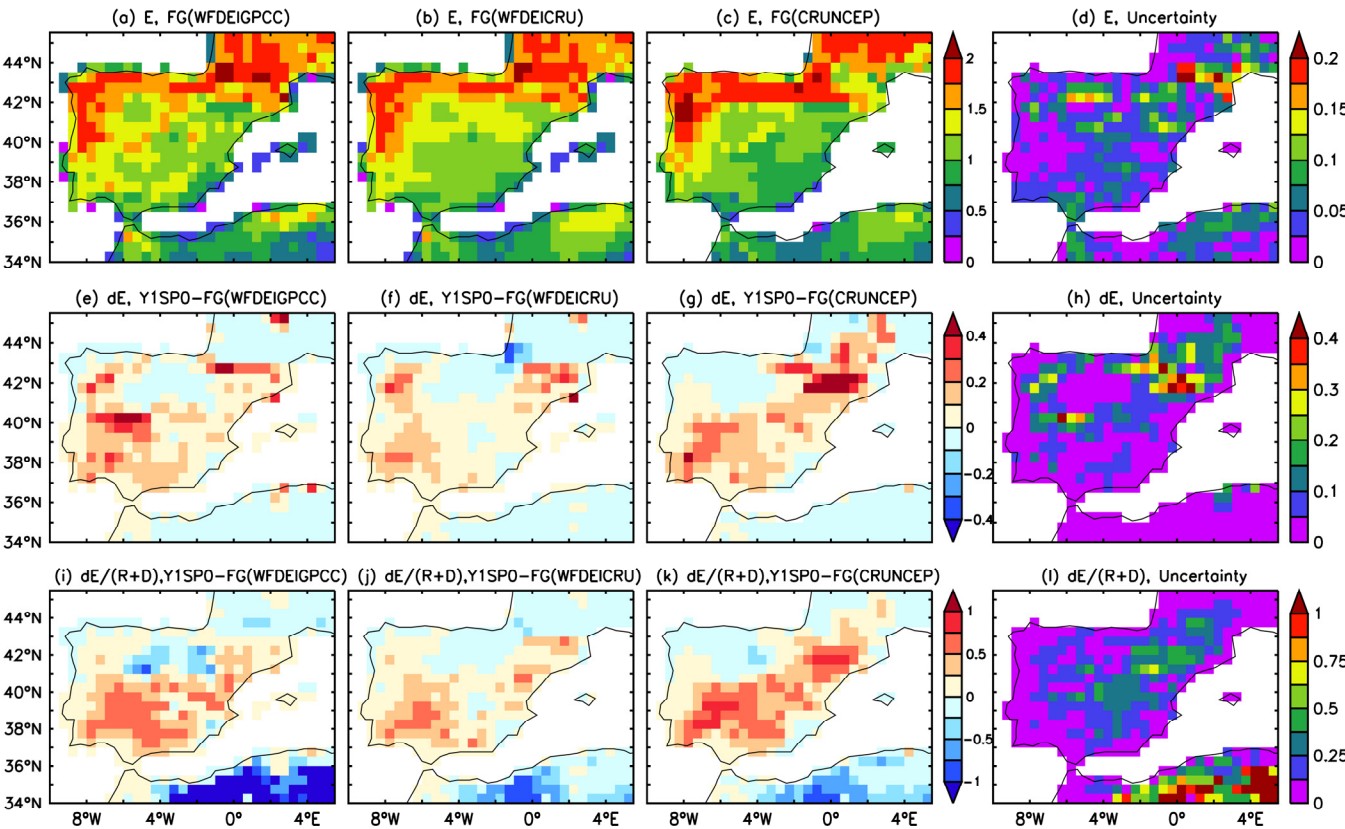

**Figure 9.** The evaporation ($E$, in mm/d) before assimilation (1st line), change of evaporation ($dE$, in mm/d) after and before assimilation (2nd line), and the ratio of $dE$ and runoff + drainage (3rd line) for forcing WFDEI-GPCC (1st column), WFDEI-CRU (2nd column), CRU-NCEP (3rd column), and the 'Uncertainty' (defined by Eq. 10) in different forcing (4th column) averaged from 1980 to 1989.

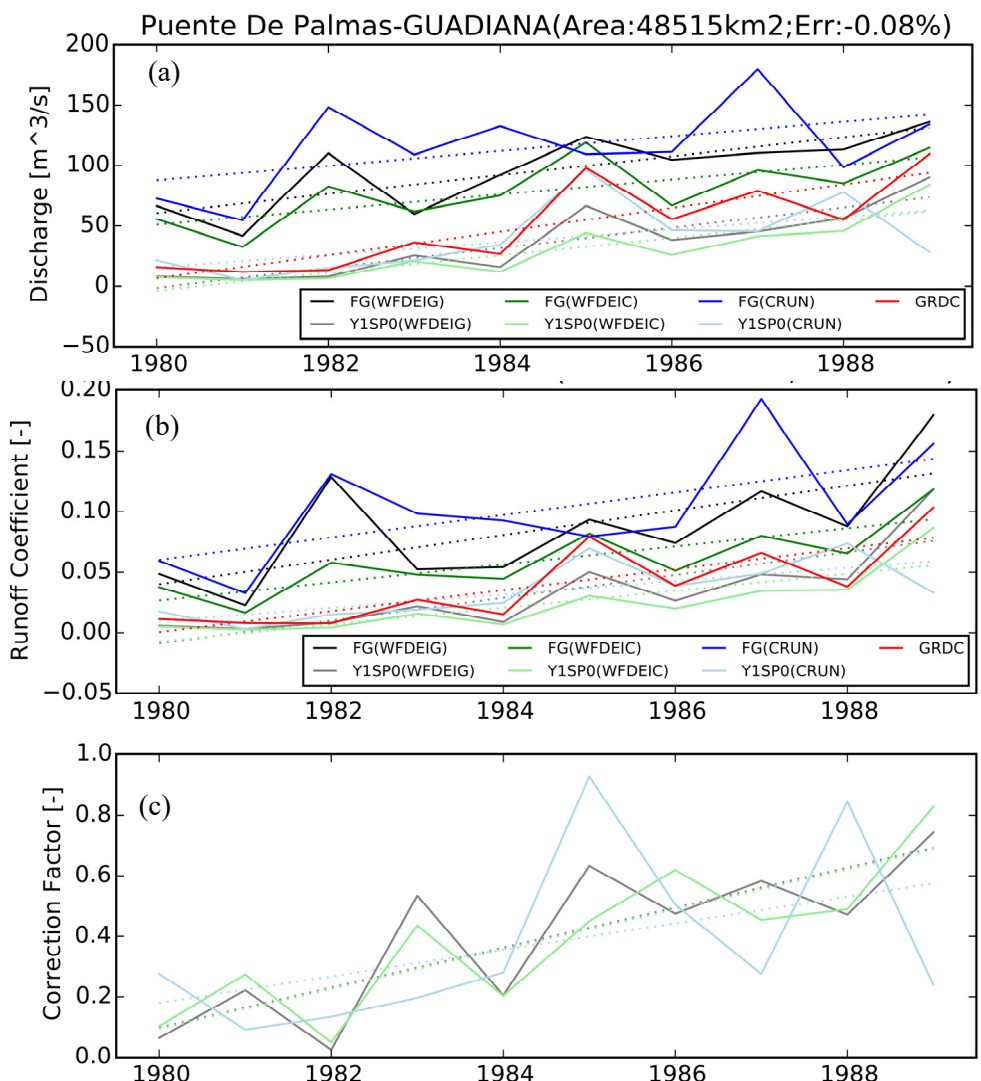

**Figure 10.** The optimization results by different atmospheric forcing (WFDEI-GPCC in black, WFDEI-CRU in green, and CRU-NCEP in blue) over the Puente De Palmas station on Guadiana River (a-d, -6.97ºW, 38.88ºN; 48515 km$^2$) and over the Masia De Pompo station on the Jucar river (e-h, -0.65ºW, 39.15ºN; 17876 km$^2$): (a, d) annual river discharges; (b, e) runoff coefficient; (e, f) optimized correction factor *x* for the simulated/assimilated river discharge (First Guess - FG in dark color, Y1SP0 in light color) with respect to GRDC observations (in red) from 1980 to 1989.

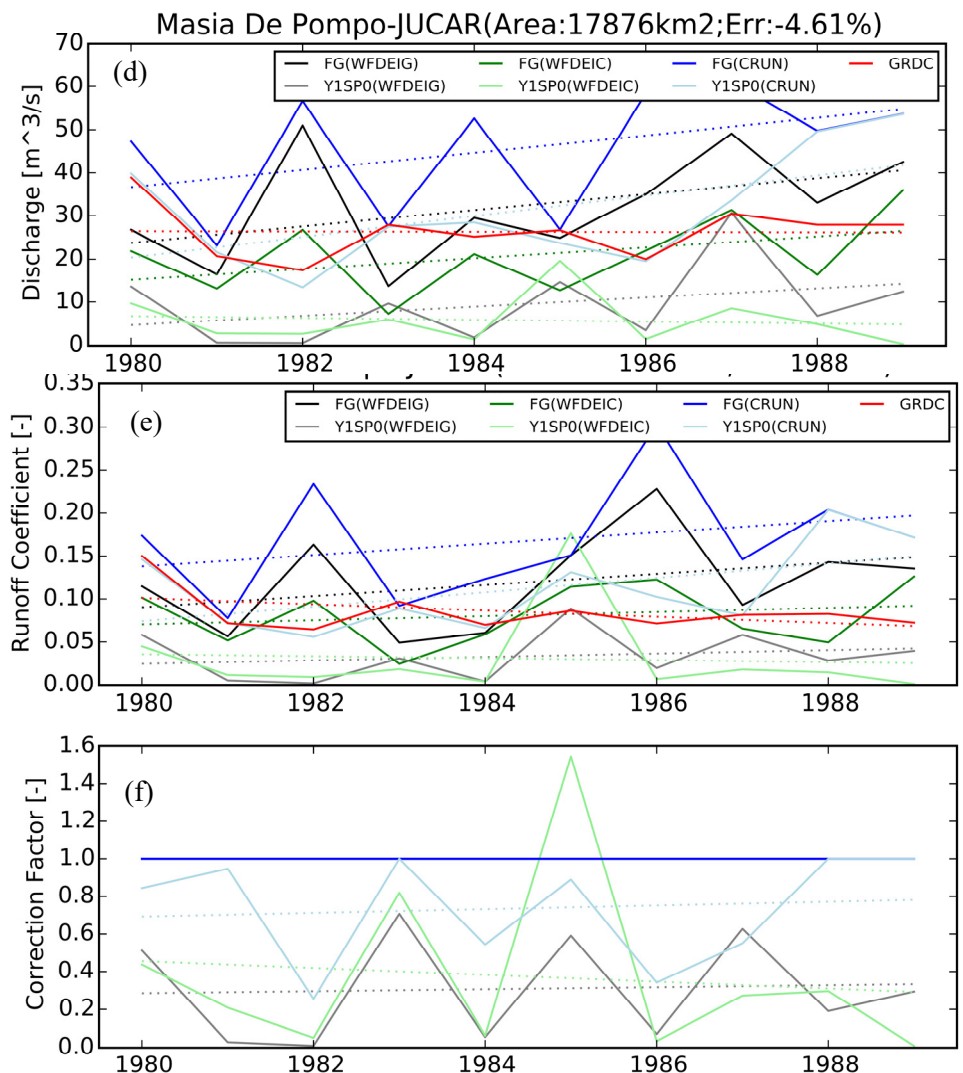

**Figure 10.** Continued.

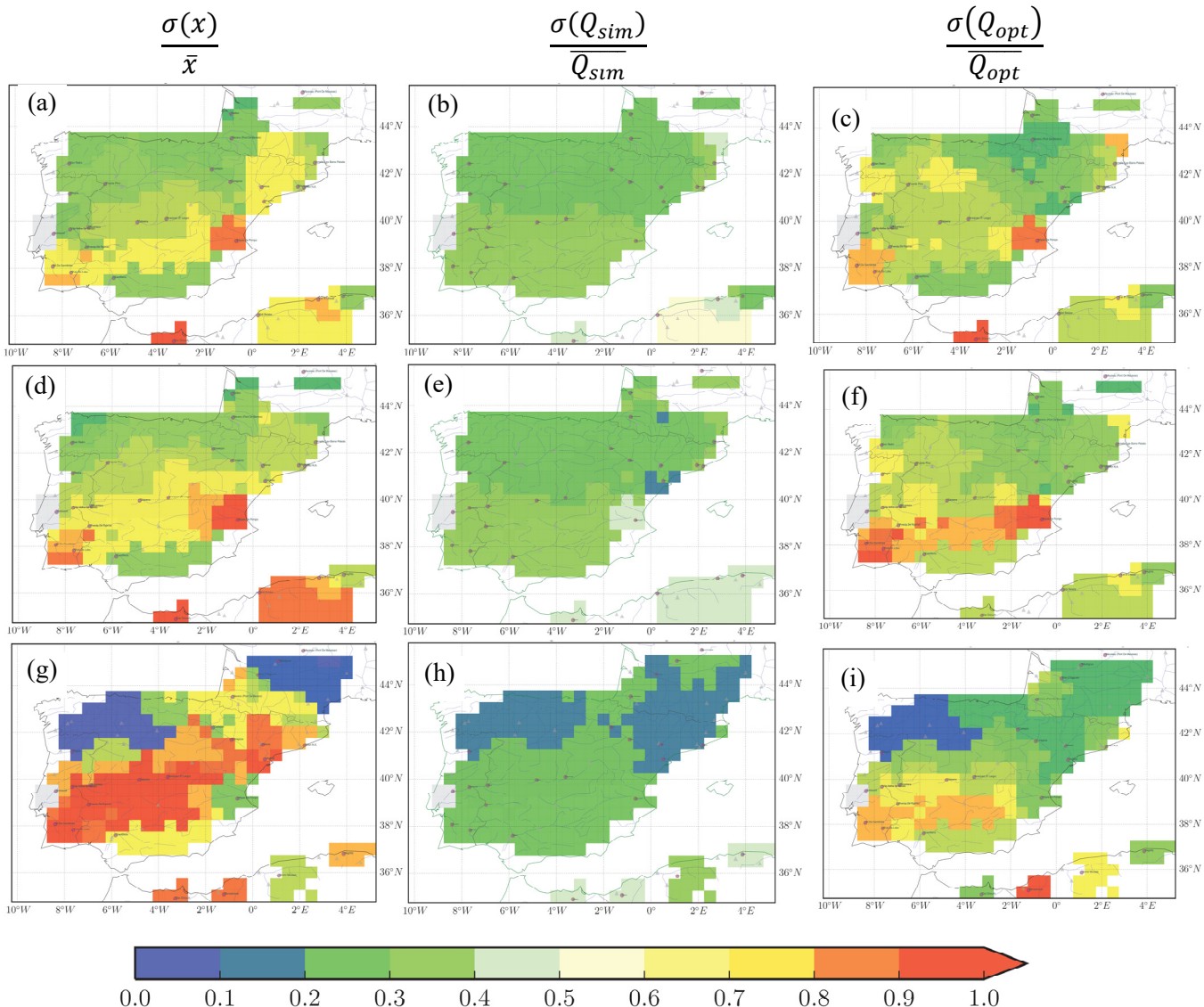

**Figure 11.** The inter-annual variation of correction factor $x$ ($\frac{\sigma(x)}{\bar{x}}$; a, d, g), simulated river discharge without assimilation ($\frac{\sigma(Q_{sim})}{\overline{Q_{sim}}}$; b, e, h) and optimized river discharge ($\frac{\sigma(Q_{opt})}{\overline{Q_{opt}}}$; c, f, i) for Y1SP0_WFDEIGPCC (1st row), Y1SP0_WFDEICRU (2nd row) and Y1SP0_CRUNCEP (3rd row) averaged over 1980-1989.

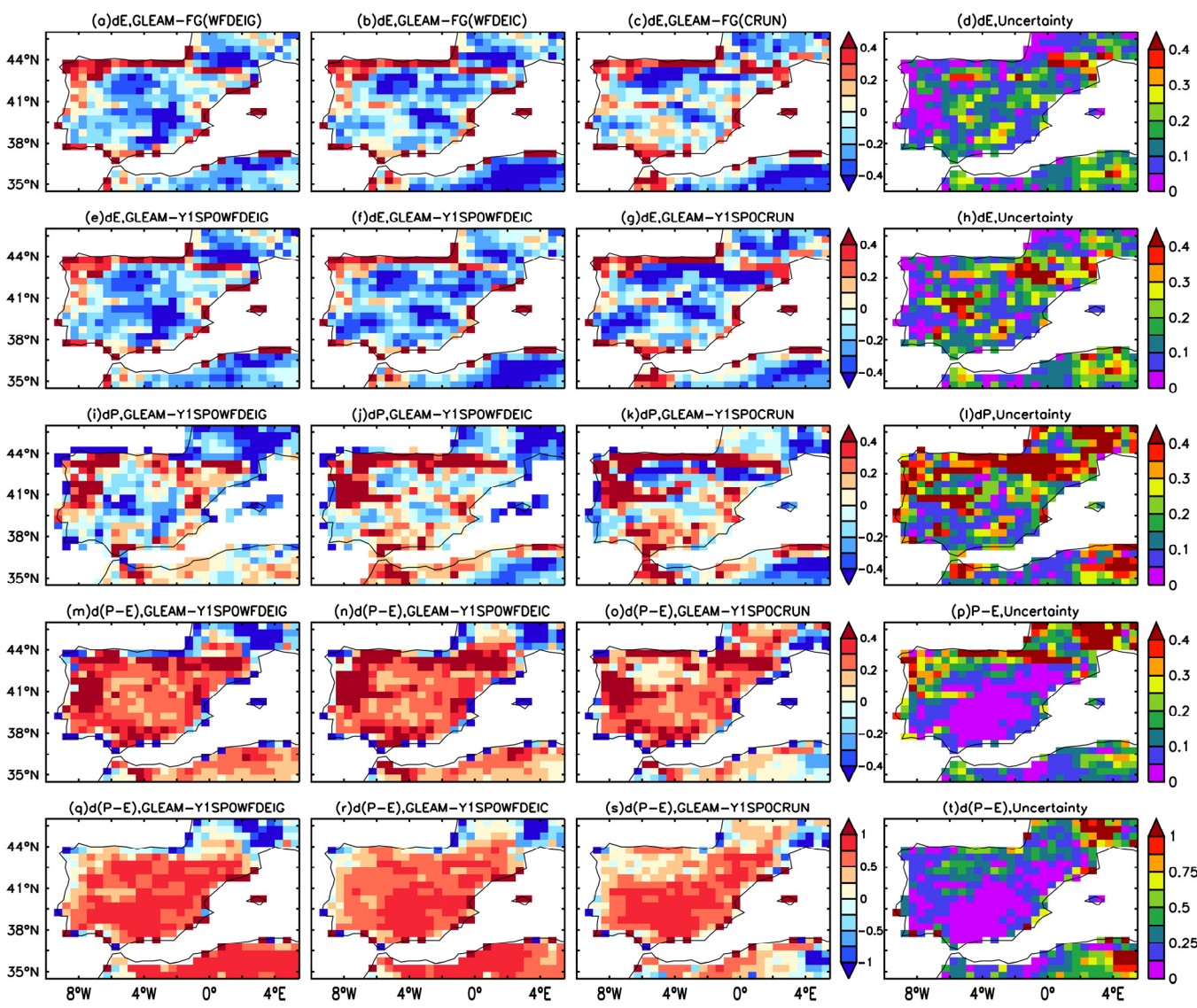

**Figure 12.** Comparison of evaporation ($E$, in mm/d, 1st line) between GLEAM (v3.1) and FG (First Guess), as well as $E$ (2nd line), precipitation ($P$, in mm/d, 3rd line), $P$-$E$ (in mm/d, 4th line) and $P$-$E$ (relative value between 0-1, 5th line) between GLEAM (v3.1) and assimilated values using different forcing (1st column: WFDEI-GPCC; 2nd column: WFDEI-CRU; 3rd column: CRU-NCEP; 4th column: 'Uncertainty' (defined by Eq. 10) of using different forcing) averaged from 1980 to 1989.

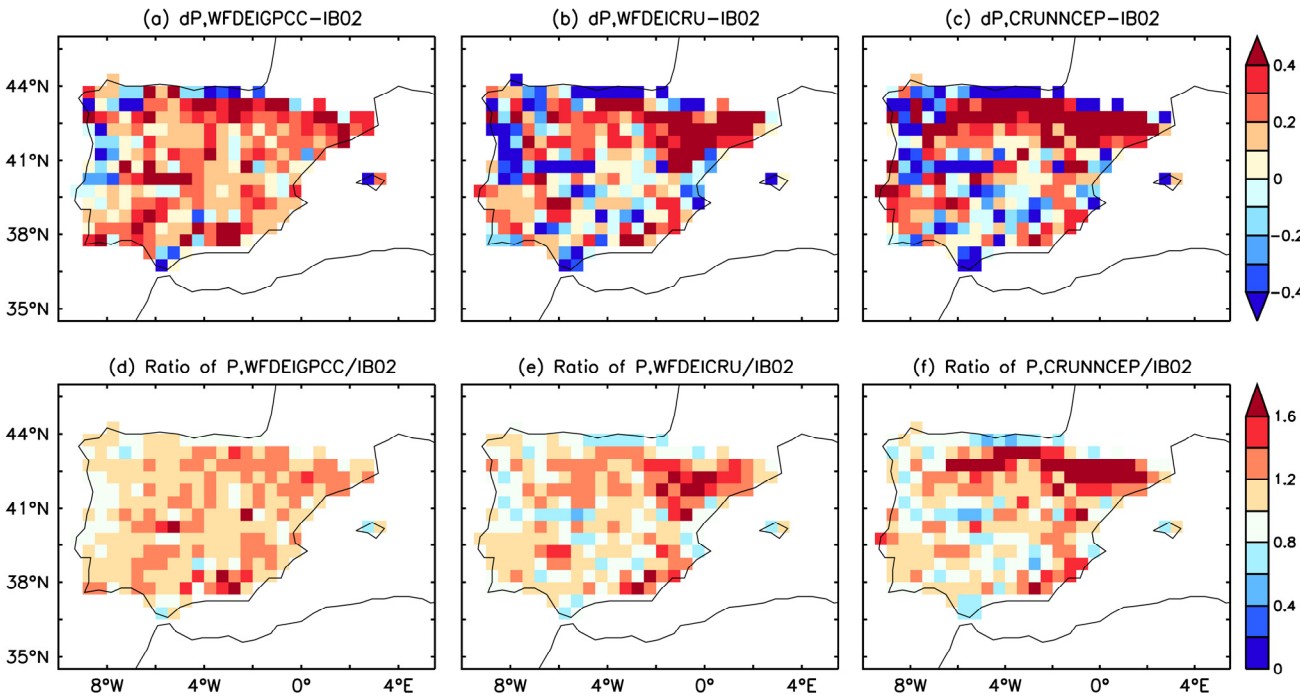

**Figure S1.** Comparison of precipitation (*P*, in mm/d) between IB02 and that used in the assimilation (a and d: WFDEI-GPCC; b and e: WFDEI-CRU; c and f: CRUNCEP) averaged from 1980 to 1989: 1[st] row for difference; 2[nd] row for ratio.