# Peer review of "Assimilation of river discharge in a land surface model to"

_Hydrology and Earth System Sciences, 2017_

## Referee Comment (RC1) · Anonymous Referee #1 · 27 Feb 2018

The paper describes a number of experiments assimilating GRDC runoff data into the ORCHIDEE land surface model across the Iberian peninsula. The assimilation adjusts the simulated runoff at sub-catchment scale where GRDC observations are available through a 'optimization parameter', effectively rescaling the simulated runoff towards the observations. The discharge bias is substantially reduced by adjusting this 'optimization parameter' and neighbouring sub-catchments are corrected by extrapolating the parameter to these.

The paper is clear and it is well written. The study is likely to be very relevant for future studies possibly extending and improving on the presented concept.

[Figure]

I propose a minor revision.

Some general remarks: The validation is performed on the basis of using GRDC both as an 'observation' and an independent 'validation' dataset? This should be discussed very critically. I am not an expert in the field of continental runoff and possibly there is no other independent data source to have a better independent validation. In general this is however quite uncommon in assimilation studies, i.e. satellite observations might be assimilated to improve soil moisture and the results would be validated against independent in-situ measurements.

For probably this reason the authors compare the corrected evaporation against GLEAM. As stated, GLEAM uses a different precipitation, the entire comparison therefore is challenging. Did the authors consider using the same precipitation as input for their experiments? It should be quite simple. Also, corrected evapotranspiration values could be compared to Fluxnet in-situ measurements. This should be either included or a strong case should be made why this was not done. The motivation of exactly / only using GLEAM should also be well presented. There are a number of alternative evapotranspiration products.

The correction factor x is applied to each sub-catchment for runoff. It was not quite clear to me how the evapotanspiration was then corrected, presumably at a grid cell level? This duality between correcting at catchment scale but the model essentially being a distributed one computing the water balance at each grid cell should be made clearer. The model runoff is corrected as it was a lumped conceptual land surface model but the relationship between this and the land surface heterogeneity is not clear to me. Also, is equifinality a serious issue? I suppose a number of optimized x can result in the same or very similar runoff downstream? Can this be mitigated by also looking at the correct seasonality of the generated runoff?

The proposed method is supposed to be superior to more simple water-balance methods? Can this be somehow quantified?

[Figure]

Despite the in general high-level language there are a number of inaccuracies (for instance missing articles).

Specific:

L155: . . . for different parameters . . ., parameters includes also variables, such as soil moisture, runoff etc.?

L172: Again, I'm getting confused with parameter and variable, I suppose parameter is x, but the actual runoff is a variable? Please take care with this throughout the text.

L173: The background error B is vital in DA, why was it chosen like this? More detail needed.

L218: Is WFDEI not being updated? Please recheck.

L237: Does each HTU have it's own location within a grid cell? Or is it more 'conceptual'. Might be helpful to clarify this in the model description. I'm assuming that they have a fixed location within each grid cell.

L266: What is meant by one optimization parameter? In my understanding the algorithm only perturbs x to find the optimum fit between the runoff simulations and observations? The river routing parameters are perturbed? Or does it depend on the number of upstream catchments with a separate x?. Not quite clear to me.

L274: . . . value '1' and a 'pre-estimated error': 'and' should be 'or'?

L282: the cost function is lower? The value of the cost function? Section Experiments design needs to be a bit clearer.

L288: factor m corresponds to number of GRDC stations?

L304: The river routing model runs at each grid cell? The distributed nature of the river routing model is not quite clear.

L322: higher than a factor of 1.5?

L359: "Summary" seems misnamed for the amount of text following

L369: They most certainly do...

L375: → can allow, remove 'of'

L377: → patterns, some inaccuracies in this area

L383: Is it also connected to topographic or other land surface features which might be not well presented by the forcing data or the model itself? Just wondering.

L475: GREAM → GLEAM

L479: references, also maybe mention more global attempts to create gridded runoff data? (can be in the introduction).

L507: Throughout the paper most errors are attributed to the lack of human influences. For sure other factors also play a large role?

Figure3 top: With the logarithmic scaling the lines mostly seem pretty horizontal. Is there a clearly visible gradient when using a different scale? Maybe add this as a window. Missing unit for J?

---

## Referee Comment (RC2) · Anonymous Referee #2 · 16 Apr 2018

Review of "Assimilation of river discharge in a land surface model to improve estimates of the continental water cycles" by Fuxing Wang and co-authors.

The manuscript presents a calibration methodology to optimize a multiplicative factor on modeled surface runoff and deep drainage using river discharge observations. The study focus over Iberia using the ORCHIDEE land surface model, incorporating a river routing scheme and benefiting from the ORCHIDEE data assimilation system. This study is of general interest for the land surface and large-scale hydrological communities presenting a novel optimization/calibration methodology. The manuscript is well presented and organized, but there are a few points that require further attention before

publication.

Comments:

1. "Data assimilation": Data assimilation is normally associated with an "update" of the model state, e.g via improved initial condition. In this study, merging modelled river discharge with observations is used to "obtain optimized discharge over the entire basin" (as mentioned in the abstract). Therefore I fell that the term "data assimilation" could be a bit misleading for the audience, since this manuscript shows a model optimization or calibration. I suggest that the authors make this point very clear to avoid confusion.

2. River routing model: Since both references of the routing model are not published yet (Nguyen-Quang et al., 2017; Zhou et al., 2017) and this is a key component of this study it is important to have a bit more details on how the three linear reservoir are represented and which model parameters are used and were defined (e.g water residence time). For example the aquifer level is referred later in the text due to spin-up, but it is not clear from the model description how the aquifers are represented in the model.

3. How does the simple estimate of the correction factor used as prior ("xprior" compares with optimized values in figure 6 ? Are the changes significant for example in terms of improved correlation?

4. Role of forcing: To discard the role of precipitation forcing, the three datasets could be compared with a high resolution precipitation dataset (IB02, Belo-Pereira et al. 2011) also in terms of mean ratios : GPCC/IB02 CRU/IB02 NCEP/IB02 and compared with the "x" correction factor. I don't see this as mandatory for the paper's publication, but would make the results more robust.

5. Impact on evaporation: Section 3.4 compares the first guess evaporation by the land-surface model with the changes in evaporation resulting for the correction as a post-processing. Would it be possible to re-run the LSM applying just a constant correction factor to evaporation ? I understand that this might be difficult to do while conserving energy, but even if energy is not conserved, it could show the impact of "improving" evaporation, that would then be reflected directly in R & D and should, in principle improve the discharge simulations.

6. Comparison with GLEAM: It would be beneficial to also present the comparison between the original Evaporation and GLEAM in addition to the results in fig.12 (could be an extra panel). Considering the results shown, I find if difficult to understand the sentence " ln 473: "This result further confirms that . . .. And some processes are probably missing in GREAM v3.1" Please expand on this discussion to clarify the basis for this assumption.

Details:

Ln 21: "earth's water cycle"

Ln 324: The relative bias shown in figure 5 highlight the biases in the South since the absolute values are low. The absolute biases might be higher in the northern areas.

Ln 351: Should be: "Fig. 7 shows the annual mean" and not "annual cycle"

Ln 357: Looking that the stations distribution in Figure 2, the station Alcala Del Rio looks very close to Cantillana. If this is the case, the good results in Alcala Del Rio might be just a direct effect of the use of Cantillana observations, and it does not "validate the hypothesis that x is distributed homogeneously over the upstream basin". Please provide the distance between the stations and difference in upstream area and mean Qobs to show that Alcala Del Rio has other tributaries than just Cantillana to justify this sentence.

Ln 429: It is not clear the the simulations "underestimate the inter-annual variability". Could you provide the standard-deviation of the annual means of the observations and simulations?

Ln 436 (results in fig 10): If we assume that the increase in discharge is due to an

increase of groundwater abstraction should we expect decrease of the correction factor since this is a process which is not represented in the model? The opposite sign with an increase of the correction factor, with higher corrections in in 1980 (around 0.2) and lower in 1989 (around 0.6) suggests that the correction factor is correcting for other processes and not human intervention? I think this is worth some discussion.

References Belo-Pereira M, Dutra E, Viterbo P. Evaluation of global precipitation data sets over the Iberian Peninsula. Journal of Geophysical Research-Earth Surface. 2011;116: D20101. doi:10.1029/2010jd015481

---

## Author Comment (AC3) · 21 May 2018

The manuscript presents a calibration methodology to optimize a multiplicative factor on modeled surface runoff and deep drainage using river discharge observations. The study focus over Iberia using the ORCHIDEE land surface model, incorporating a river routing scheme and benefiting from the ORCHIDEE data assimilation system. This study is of general interest for the land surface and large-scale hydrological communities presenting a novel optimization/calibration methodology. The manuscript is well presented and organized, but there are a few points that require further attention before publication.

**Comments:**

**1.** "Data assimilation": Data assimilation is normally associated with an "update" of the model state, e.g., via improved initial condition. In this study, merging modelled river discharge with observations is used to "obtain optimized discharge over the entire basin" (as mentioned in the abstract). Therefore I fell that the term "data assimilation" could be a bit misleading for the audience, since this manuscript shows a model optimization or calibration. I suggest that the authors make this point very clear to avoid confusion.

**Answer:** The data assimilation could be applied for different cases: (1) to correct initial condition (correcting state variable) which is mostly used for numerical weather prediction; (2) to correct the state variable during the data assimilation period (i.e., in this case both the trajectory of the model and the initial conditions are corrected); (3) to correct the parameter of a model. These different usages can be mixed. In the current study, the data assimilation refers to the 3rd case which is mainly used in ORCHIDEE data assimilation and in other land surface models.

   We find similar descriptions of data assimilation in several papers. For example, Reichle (2008) mentioned that 'All data assimilation methods share the basic tenet of merging models and observations, yet the sophistication of the merging algorithm varies widely. Important differences also remain between the specific methods that are most suitable for a given application. Since atmospheric and oceanic dynamics are chaotic (that is, small errors in the initial condition can lead to large differences at later times in the model integration), data assimilation in these areas is very much concerned with the estimation of initial conditions. By contrast, land surface dynamics are damped, and land surface assimilation is all about estimating errors in uncertain meteorological forcing (boundary) conditions and model parameterizations. Clearly, ''one size does not fit all'' in data assimilation'. Smith et al. (2013) explained that 'It is most commonly used to produce initial conditions for state estimation: estimating model variables whilst keeping the model parameters fixed. However, it is also possible to use data assimilation to provide estimates of uncertain model parameters.' Raoult et al. (2016) also wrote that 'Optimisation techniques come under the umbrella of model–data fusion and range from simple ad hoc parameter tuning to rigorous data assimilation frameworks. These approaches have been used in a number of studies, covering various LSMs, to derive vectors of parameters that improve model–data fit significantly.'

For this reason, the expression of 'data assimilation' is kept in the paper, but a clarification is given at **Lines 91-97**: The data assimilation, a specific type of inverse problem, is generally applied for different cases: (1) to correct initial condition (correcting state variable) which is mostly used for numerical weather prediction; (2) to correct the state variable during the data assimilation period (i.e., in this case both the trajectory of the model and the initial conditions are corrected); and (3) to correct the parameter of a model. In the current study, the data assimilation refers to the $3^{rd}$ case.

**Answer:** For easier understanding, the methods of '$x_{prior} = 1$' and '$x_{prior} =$ pre-estimated-prior' are named as $x_{prior\_1}$ and $x_{prior\_ref}$, respectively (**Section 2.4 and Fig. 3**).

The $x_{prior\_ref}$ is compared with optimized correction factor in Fig. R1 below. The $x_{prior\_ref}$ captures the general distribution pattern of optimal $x$, but the correlation coefficient of using $x_{prior\_ref}$ is lower than that of using optimal $x$. In other words, the assimilated river discharge is improved through both choosing $x_{prior\_ref}$ and optimization. The role of optimization is to find an appropriate correction factor when there are several basins (with observations) overlaps at upstream.

[Figure]

**Figure R1.** The $x_{prior\_ref}$ (left) and the correlation coefficient (right) of river discharge between observations and simulations from 1980 to 1989 for WFDEI_GPCC (1st row), WFDEI_CRU (2nd row) and CRU_NCEP (3rd row) forcing.

Explanations were added in **Lines 397-401**: 'It should be mentioned that the $x_{prior\_ref}$ is able to capture the general distribution pattern of optimal $x$, but the performance of river discharge estimation is significantly improved through optimization. The role of optimization is to find an appropriate correction factor when there are several basins (with observations) overlaps at upstream'.

**4.** Role of forcing: To discard the role of precipitation forcing, the three datasets could be compared with a high resolution precipitation dataset (IB02, Belo-Pereira et al. 2011) also in terms of mean ratios: GPCC/IB02 CRU/IB02 NCEP/IB02 and compared with the "$x$" correction factor. I don't see this as mandatory for the paper's publication, but would make the results more robust.

Belo-Pereira M, Dutra E, Viterbo P. Evaluation of global precipitation data sets over the Iberian Peninsula. Journal of Geophysical Research-Earth Surface. 2011. 116: D20101. doi:10.1029/2010jd015481.

**Answer:** The precipitation of WFDEI_GPCC, WFDEI_CRU, and CRU_NCEP is compared with the IB02 precipitation data. The precipitation of the three forcing are higher than IB02 over most regions (Figs. R2a-R2c) but their spatial distributions are different with the proposed evaporation correction (Figs. 9e-9g). The ratios of WFDEI_GPCC/IB02, WFDEI_CRU/IB02 CRU_NCEP/IB02 are generally higher than 1 with few grid cells of ratios lower than 1 being distributed randomly (Figs. R2d-R2f). The pattern of the three ratios is not consistent with the optimized correction factor (Figs. 8a-8c), which indicates that the precipitation forcing error is not likely the dominant factor of the correction factor distribution.

These analysis are added in the revised manuscript (**Lines 452-459**): 'This is also demonstrated by comparing the precipitations between the three forcing and IB02 dataset. Compared to IB02, all the three forcing overestimate rainfall in the Iberian Peninsula (Figs. S1a-S1c), but none of these error patterns resembles that of the proposed $E$ correction (Figs. 9e-9g). Unlike the pattern of the correction factor (Figs. 8a-8c), the ratios of annual mean precipitation between the three forcing and IB02 are higher than 1 over most regions (Figs. S1d-S1f). Therefore, the precipitation forcing error is not likely the dominant factor in determining the correction factor distribution.'

The IB02 dataset is described at **Lines 254-257**: 'The precipitation of the three forcing is compared with IB02 dataset which is a gridded daily rainfall dataset for Iberia Peninsula with 0.2° resolution covers 1950 to 2003 (Belo-Pereira et al., 2011). It is generated by using ordinary kriging from more than 2400 quality-controlled stations.'

The reference was added at **Lines 650-652.** The Figs. R2d-R2f below were added in the '**Supplementary**' of the manuscript (Fig. S1).

[Figure]

**Figure R2.** Comparison of precipitation ($P$, in mm/d) between IB02 and that used in the assimilation (a and d: WFDEI-GPCC; b and e: WFDEI-CRU; c and f: CRUNCEP) averaged from 1980 to 1989: 1st row for difference; 2nd row for ratio.

**5.** Impact on evaporation: Section 3.4 compares the first guess evaporation by the land-surface model with the changes in evaporation resulting for the correction as a post-processing. Would it be possible to re-run the LSM applying just a constant correction factor to evaporation ? I understand that this might be difficult to do while conserving energy, but even if energy is not conserved, it could show the impact of "improving" evaporation, that would then be reflected directly in R & D and should, in principle improve the discharge simulations.

**Answer:** We tested the possibility of improving river discharge by using a constant correction factor to evaporation. Theoretically, the modification of evaporation leads to a change in soil moisture thus surface runoff and deep drainage are changed. From Eq. (6), the correction factor for $E$ ($X_{Ecorr}$) can be derived from $x$ by Eq. (R1). The $X_{Ecorr}$ is then applied to correct $E$ (Eq. R2). Like the correction factor $x$, the $X_{Ecorr}$ changes with year.

$$X_{Ecorr} \approx \frac{E + (1 - x) \cdot (R + D)}{E}, \qquad (R1)$$

$$E_{corr} = X_{Ecorr} \cdot E \qquad (R2)$$

The Eqs. R1 and R2 were implemented in ORCHIDEE LSM and the LSM was running over 1980-1984. The Fig. R3 shows the BIAS of river discharge after correcting evaporation in ORCHIDEE

LSM. The absolute BIAS is reduced comparing with the reference run (forced by WFDEI_GPCC without correcting evaporation).

The BIAS becomes negative after correcting evaporation, which is probably because the evaporation correction factor $X_{Ecorr}$ is greater than 1 over most cases, and it leads to a decrease in R+D with time evolution. Unlike the correction of runoff and drainage by using a constant factor (change with year) in current study, the correction of evaporation leads to a feedback on soil moisture which in turn affects the evaporation simulation. Therefore, both energy and water balance are not conserved in this case. Another solution of improving river discharge simulation by correcting evaporation could be to run the full ORCHIDEE LSM in the assimilation system with the same cost function as Eq. (7) in the manuscript. In this way, the intermediate variables are adjusted towards optimal river discharge with the modification of evaporation. Because the optimization by running the full ORCHIDEE model is very time consuming, this is not done in this paper but could be one of the future work.

The explanations were added in the revised manuscript (**Line 487-496**): 'We also tested the possibility of improving the river discharge estimation by using a constant correction factor to evaporation ($X_{Ecorr}$). The $X_{Ecorr}$ (different for each year) can be derived from Eq. (6).

$$X_{Ecorr} \approx \frac{E + (1 - x) \cdot (R + D)}{E},\tag{11}$$

$$E_{corr} = X_{Ecorr} \cdot E\tag{12}$$

Although the Eqs. 11-12 are able to improve river discharge estimation by modifying soil moisture, the energy and water balance are not conserved. One solution could be to run the full ORCHIDEE LSM in the assimilation system with the same cost function as Eq. (7). In this way, the intermediate variables are adjusted towards optimal river discharge with the modification of evaporation. This approach executes the full ORCHIDEE model thus is very time consuming and is beyond the scope of the current study.'

[Figure]

**Figure R3.** The BIAS of simulated river discharge before (left) and after (right) correcting evaporation in LSM by correction factor from 1980 to 1984.

**6.** Comparison with GLEAM: It would be beneficial to also present the comparison between the original Evaporation and GLEAM in addition to the results in Fig. 12 (could be an extra panel). Considering the results shown, I find it difficult to understand the sentence " ln 473: "This result further confirms that ….. And some processes are probably missing in GREAM v3.1". Please expand on this discussion to clarify the basis for this assumption.

**Answer:** The comparison between the original Evaporation and GLEAM is shown in Fig. S4 below (and **Figs. 12a-12d** in the revised manuscript).

The explanations have been added **at Lines 551-554:** 'We find large difference between GLEAM and FG, which indicates that the evaporation is quite uncertain for different estimations. The geographical distribution and magnitude of difference in $E$ between GLEAM and FG is highly consistent with that between GLEAM and bias corrected values by using different forcing (Figs. 12a-12c, and 12e-12g).'

The sentences have been revised at **Lines 558-561** to avoid confusion: 'Because the bias corrected $P$-$E$ are corrected by GRDC observed river discharge, the $P$-$E$ (≈river discharge) of GLEAM is very likely to be higher than GRDC observations over the Iberia. This result indicates that some processes are probably also missing in GLEAM v3.1.'

[Figure]

**Figure R4.** Comparison of evaporation (*E*, in mm/d) between GLEAM (v3.1) and FG values using different forcing (a: WFDEI-GPCC; b: WFDEI-CRU; c: CRUNCEP; d: uncertainty of using different forcing) averaged from 1980 to 1989.

**Details:**

**1.** Ln 21: "earth's water cycle"

**Answer:** Revised (**Line 21**).

**2.** Ln 324: The relative bias shown in figure 5 highlight the biases in the South since the absolute values are low. The absolute biases might be higher in the northern areas.

**Answer:** Fig. R5 below plots the absolute bias over 1980-1989 by three different forcing. The high values of absolute bias are distributed in both northern and southern areas, and its spatial distribution is different for different forcing. To avoid confusion, the BIAS is named normalized bias (*Norm_BIAS*) in the revised manuscript.

The explanations have been added at **Lines 369-371**: 'The spatial pattern of the absolute bias in river discharge varies with the atmospheric forcing (not shown). The normalized bias is then applied to measure the river discharge simulation.

The expression was also revised at **Line 374**: 'The *Norm_BIAS* is small (within +/- 0.3) over north, west and southeast of the region (Figs. 5b, 5d and 5f)'.

[Figure]

**Figure R5.** The absolute bias (m³/s) of river discharge simulations from 1980 to 1989 using WFDEI_GPCC (a), WFDEI_CRU (b) and CRU_NCEP (c) forcing.

**3.** Ln 351: Should be: "Fig. 7 shows the annual mean" and not "annual cycle"

**Answer:** Revised to 'annual mean' (**Line 407**).

**4.** Ln 357: Looking that the stations distribution in Figure 2, the station Alcala Del Rio looks very close to Cantillana. If this is the case, the good results in Alcala Del Rio might be just a direct effect of the use of Cantillana observations, and it does not "validate the hypothesis that $x$ is distributed homogeneously over the upstream basin". Please provide the distance between the stations and difference in upstream area and mean $Q_{obs}$ to show that Alcala Del Rio has other tributaries than just Cantillana to justify this sentence.

**Answer:** Based on GRDC observations, the distance and the difference in upstream area between Alcala Del Rio and Cantillana stations are 15.3 km and 2124 km² (46995 km² and 44871 km², respectively). Between the two stations, there are several tributaries flow to Alcala Del Rio station, which leads to different annual mean river discharges at Cantillana (49.7 m³/y) and Alcala Del Rio stations (94.8 m$^3$/y). This result illustrates that this approach is able to correct the river discharge over the entire basin. The above numbers and expressions were provided in the revised manuscript at **Lines 409-416**: 'The observation of this station is not assimilated due to its large upstream area difference (15.53%>10%) between model (55635 km$^2$) and GRDC (46995 km$^2$). The overestimated discharge simulated by the model at this station is also corrected because it benefits from the correction factor estimated at the Cantillana station (-5.83ºW, 37.59ºN; 44871 km$^2$) which locates at the 15.3 km upstream of Alcala Del Rio station of the Guadalquivir River (southwest of the Iberian Peninsula). Between the two stations, there are several tributaries flow to Alcala Del Rio station, which leads to different annual mean river discharges at Cantillana (49.7 m$^3$/y) and Alcala Del Rio stations (94.8 m$^3$/y). This result illustrates that this approach is able to correct the river discharge over the entire basin.'

**5.** Ln 429: It is not clear that the simulations "underestimate the inter-annual variability". Could you provide the standard-deviation of the annual means of the observations and simulations?

**Answer:** The standard-deviations of the annual means for the FG(WFDEIG) and FG(WFDEIC) are 28.8 m$^3$/s and 25.2 m$^3$/s, respectively. They are lower than observation (33.8 m$^3$/s). The values are provided at **Line 506-509**: "… while the FG(WFDEIG) and FG(WFDEIC) underestimate the inter-annual variability comparing with observations (Fig. 10a-10b). The standard-deviation of the annual means for observation, FG(WFDEIG), FG(WFDEIC) and FG(CRUN) are 33.8 m$^3$/s, 28.8 m$^3$/s, 25.2 m$^3$/s and 34.3 m$^3$/s, respectively."

**6.** Ln 436 (results in Fig. 10): If we assume that the increase in discharge is due to an increase of groundwater abstraction should we expect decrease of the correction factor since this is a process which is not represented in the model? The opposite sign with an increase of the correction factor, with higher corrections in 1980 (around 0.2) and lower in 1989 (around 0.6) suggests that the correction factor is correcting for other processes and not human intervention? I think this is worth some discussion.

**Answer:** The following sentences and references were removed to avoid confusion: "
[revised manuscript text omitted]

---

## Author Comment (AC4) · 24 May 2018

*The paper describes a number of experiments assimilating GRDC runoff data into the ORCHIDEE land surface model across the Iberian peninsula. The assimilation adjusts the simulated runoff at sub-catchment scale where GRDC observations are available through a 'optimization parameter', effectively rescaling the simulated runoff towards the observations. The discharge bias is substantially reduced by adjusting this 'optimization parameter' and neighboring sub-catchments are corrected by extrapolating the parameter to these.*

*The paper is clear and it is well written. The study is likely to be very relevant for future studies possibly extending and improving on the presented concept.*

*I propose a minor revision.*

**Some General Remarks:**

**1.** *The validation is performed on the basis of using GRDC both as an 'observation' and an independent 'validation' dataset? This should be discussed very critically. I am not an expert in the field of continental runoff and possibly there is no other independent data source to have a better independent validation. In general this is however quite uncommon in assimilation studies, i.e. satellite observations might be assimilated to improve soil moisture and the results would be validated against independent in-situ measurements.*

**Answer:** Yes, the GRDC is used for both river discharge assimilation and validation due to the fact that this is the only comprehensive dataset at global scale so far. To overcome this limitation, the assimilated river discharges are also validated over the catchments where the GRDC stations are discarded during assimilation (e.g., Alcala Del Rio station of the Guadalquivir River in Fig. 7). Although the validation datasets are from the same GRDC source, they are from other independent observation stations thus can be seen as an independent validation. This is a 'first order validation' since the stations are discarded from the assimilation, but we expect the first order is valid.

The explanations are added in **Lines 402-406**: 'A common validation approach is to compare the assimilated river discharge with other independent data sources. However, the river discharge observations are limited, and the GRDC is the only comprehensive river discharge datasets at global scale so far. To overcome this limitation, the assimilated river discharges are also validated over the catchments where the GRDC stations are discarded during assimilation.'

And in **Lines 416-420**: 'The discharges for certain sub-basins without assimilated observations (e.g., observation unavailable or GRDC stations discarded) are corrected by $x$ as well. Although the validation datasets are from the same GRDC source, they are from other independent observation stations thus can be seen as an independent validation ('first order validation').

**2. (1)** *For probably this reason the authors compare the corrected evaporation against GLEAM. As stated, GLEAM uses a different precipitation, the entire comparison therefore is challenging. Did the authors consider using the same precipitation as input for their experiments? It should be quite simple.*

**Answer:** The experiments of using the GLEAM's precipitation as input are not carried out due to three reasons. First, parts of the atmospheric forcing (e.g., air pressure, air humidity, wind speed, etc.) of GLEAM are not available, and they also impact evapotranspiration estimation. Second, it is not feasible to use GLEAM precipitation and maintaining a coherence with other forcing (e.g. radiation) taken from other sources. Third, the assimilation system is run with three different atmospheric forcing (WFDEI_GPCC, WFDEI_CRU, CRU_NCEP, see Fig. 12a-12d), and the evapotranspiration corrections are quite close (see Fig. 12a-12d). The explanations were added in **Lines 548-551:** 'Due to the unavailability of parts of GLEAM's atmospheric forcing (e.g., air pressure, air humidity, air speed, etc.) and difficulty of maintaining a coherence with other forcing, the assimilation system does not run with GLEAM's precipitation input'.

**2. (2)** *Also, corrected evapotranspiration values could be compared to Fluxnet in-situ measurements. This should be either included or a strong case should be made why this was not done. The motivation of exactly / only using GLEAM should also be well presented. There are a number of alternative evapotranspiration products.*

**Answer:** The corrected evapotranspiration ($E$) is compared with several other evapotranspiration products (Fig. R1 below). The 'FluxNex extrapolated' dataset (1982-2008, at 0.5°) was obtained by applying machine learning approach to the upscaling of observations from the global network of eddy covariance towers FLUXNET (Jung et al., 2009). The precipitation of MPIBGC for model tree training comes from CRU. Vinukollu et al. (2011) generated global evaporation from multi-sensor remote sensing data using Penman-Monteith and Priestley-Taylor based approach ('PUPM' and 'PUPT' hereafter). This data is available at 0.5° from 1984 to 2007 with its precipitation forcing obtained from GPCP. Mueller et al. (2013) collected most of the existing evaporation products and produced

LandfluxEVAL from 1989 to 2005 at 1°. The 'Diagnostic' and 'All' categories are chosen to use as much observations as possible. These comparison results are consistent with the comparison with GLEAM.

The availability of both precipitation and $E$ for GLEAM data allows to estimate $P$-$E$ which can be compared with the assimilate values, while the $P$-$E$ is not available for other datasets. Therefore, only the results of GLEAM are shown in the study.

[Figure]

**Figure R1.** Comparison of assimilated $E$ (mm/d) forced by WFDEI_GPCC with other products: PUPM (a), PUPT (b), LandfluxEVALAll (c), LandfluxEVALDiag (d), and 'FluxNet extrapolated' (e) by using multi-year averaged values from 1980 to 1989.

The explanations are added (**Lines 563-567**): 'The results are quite consistent when comparing the corrected $E$ with several other products which are obtained by using different methodology and forcing (e.g., Jung et al., 2009; Vinukollu et al., 2011; Mueller et al., 2013). Considering the availability of $P$-$E$ for GLEAM data which allows to compare it with the bias corrected value, only the results of GLEAM are shown'. Three new references were added (**Lines 701-703, 752-756, 838-840**):

Jung, M., Reichstein, M., and Bondeau, A.: Towards global empirical upscaling of FLUXNET eddy covariance observations: validation of a model tree ensemble approach using a biosphere model, Biogeosciences, 6, 2001–2013, doi:10.5194/bg-6-2001-2009, 2009.

Mueller, B., Hirschi, M., Jimenez, C., Ciais, P., Dirmeyer, P. A., Dolman, A. J., Fisher, J. B., Jung, M., Ludwig, F., Maignan, F., Miralles, D., McCabe, M. F., Reichstein, M., Sheffield, J., Wang, K. C., Wood, E. F., Zhang, Y., and Seneviratne, S. I.: Benchmark products for land evapotranspiration: LandFlux-EVAL multi-dataset synthesis, Hydrol. Earth Syst. Sci., 17, 3707-3720, doi:10.5194/hess-17-3707-2013, 2013.

Vinulcollu, R.K., Wood, E.F., Ferguson, C.R., Fisher, J.B.: Global estimates of evapotranspiration for climate studies using multi-sensor remote sensing data: Evaluation of three process-based approaches, Remote sensing of environment, 115(3):801-23, 2011.

**3. (1)** *The correction factor x is applied to each sub-catchment for runoff. It was not quite clear to me how the evapotranspiration was then corrected, presumably at a grid cell level? This duality between correcting at catchment scale but the model essentially being a distributed one computing the water balance at each grid cell should be made clearer. The model runoff is corrected as it was a lumped conceptual land surface model but the relationship between this and the land surface heterogeneity is not clear to me.*

**Answer:** The evapotranspiration and the 'runoff + drainage' are corrected at the same grid cell level (dashed lines in Fig. 1a). Over each grid cell, the 'runoff + drainage' is corrected by $x$, while the corrected evapotranspiration is obtained from Eq. (5). Over each sub-catchment (with several model grids) where the GRDC is available, the correction factors over these model grid cells are the same. These explanations have been added at **Lines 158-159**: 'Over each upstream area (dashed box in Fig. 1a), the optimal $x$ of these model grid cells are the same. The '$R + D$' and $E$ are corrected at the same grid cell level by $x$ and Eq. (5), respectively.'

**3. (2)** *Also, is equifinality a serious issue? I suppose a number of optimized x can result in the same or very similar runoff downstream? Can this be mitigated by also looking at the correct seasonality of the generated runoff?*

**Answer:** Yes, the reviewer is right. A number of optimized $x$ can result in very similar river discharge. Due to the high requirement of computing resources, the assimilation time step is yearly instead of monthly. Using monthly data with monthly $x$ may not completely alleviate the equifinality issue. By using annual $x$ values with monthly discharge, we assume that the seasonal variations of surface runoff and deep drainage are perfect.

Explanations were added in **Lines 623-627**: 'To improve the calculation efficiency, this study uses annual mean correction factors without considering its seasonal variation thus the seasonal discharges do not improved. One issue of the $x$ optimization approach could be the equifinality with a number of optimized $x$ result in the similar river discharge at downstream.). Future developments can be made towards generating ensemble optimal *x* to better assess the uncertainties associated to each parameter *x*.'

**4.** *The proposed method is supposed to be superior to more simple water-balance methods? Can this be somehow quantified?*

**Answer:** Yes, the proposed method is supposed to be superior to simple water-balance methods, because a LSM has better estimation of evapotranspiration (*E*) using physically based equations and takes advantage of spatial distribution of precipitation (*P*) and *P-E*. Furthermore, the LSM simulates river discharge at a higher frequency (daily) than the simple water-balance methods (monthly to annual). Explanations were added in **Lines 614-617**: 'The proposed method is supposed to be superior to the simple water-balance methods, because a LSM estimates *E* at sub-diurnal scales with physically based equations and takes advantage of spatial distribution of the *P* and *P-E*'.

This is difficult to quantify because both the distributed LSM and the simple water-balance model can be adjusted with data assimilation, but we hope that the correction factor will be smaller for LSM than other water balance methods.

**5.** *Despite the in general high-level language there are a number of inaccuracies (for instance missing articles).*

**Answer**: The grammar was checked and the articles were added in several places (e.g., added 'the' before 'Iberian Peninsula', before 'Mediterranean', before 'Jucar River', before 'Chelif', before 'ESA', etc.).

**Specific comments:**

**1.** *L155: . . . for different parameters . . ., parameters includes also variables, such as soil moisture, runoff etc.?*

**Answer:** The sentence 'for various parameters' was revised to "for various variables" (**Line 166**).

**2.** *L172: Again, I'm getting confused with parameter and variable, I suppose parameter is x, but the actual runoff is a variable? Please take care with this throughout the text.*

**Answer:** Revised (**Lines 163**). The 'parameter' now refers to *x*, while others are 'variable'.

**3.** *L173: The background error B is vital in DA, why was it chosen like this? More detail needed.*

**Answer:** Details were added at **Lines 185-187**: 'The matrix **B** was determined based on expert knowledge of ORCHIDEE model (Kuppel et al., 2012; Santaren et al., 2014)'. The related references were provided (**Lines 716-718, 799-802**):

Kuppel, S., Peylin, P., Chevallier, F., Bacour, C., Maignan, F., and Richardson, A. D.: Constraining a global ecosystem model with multi-site eddy-covariance data, Biogeosciences, 9, 3757-3776, https://doi.org/10.5194/bg-9-3757-2012, 2012.

Santaren, D., Peylin, P., Bacour, C., Ciais, P., and Longdoz, B.: Ecosystem model optimization using in situ flux observations: benefit of Monte Carlo versus variational schemes and analyses of the year-to-year model performances, Biogeosciences, 11, 7137-7158, doi:10.5194/bg-11-7137-2014, 2014.

**4.** *L218: Is WFDEI not being updated? Please recheck.*

**Answer:** Yes, the WFDEI is updated with time (revised at **Line 248**).

**5.** *L237: Does each HTU have it's own location within a grid cell? Or is it more 'conceptual'. Might be helpful to clarify this in the model description. I'm assuming that they have a fixed location within each grid cell.*

**Answer:** The location of each HTU is not fixed within each grid cell. Each river basin is constructed by connecting a number of HTUs which are defined inside the ORCHIDEE grid boxes. Each HTU represents the section of the river basin within the grid box. Therefore, the location of each HTU within the grid cell is not fixed but depends on basin characters.

The explanations were added at **Lines 208-211: '**The HTU is constructed based on the Pfafstetter topological coding system and user defined size. Each HTU represents the section of the river basin within the grid box, and many HTUs forms a river basin (Nguyen-Quang et al., 2018). Therefore, the relative locations of HTUs in each grid cell are not fixed**'.**

More details of HTU are elaborated by Nguyen-Quang et al. (2018, **Lines 766-769**):

Nguyen-Quang T., Polcher, J., Ducharne, A., Arsouze. T., Zhou X., Schneider A., and Fita L.: ORCHIDEE-ROUTING: A new river routing scheme using a high resolution hydrological database. Submitted to Geosci. Model Dev., 2018

**6.** *L266: What is meant by one optimization parameter? In my understanding the algorithm only perturbs x to find the optimum fit between the runoff simulations and observations? The river routing parameters are perturbed? Or does it depend on the number of upstream catchments with a separate x?. Not quite clear to me.*

**Answer:** It means to perturb $x$ over one upstream catchment in each iteration, but not perturbing the river routing parameters. Different upstream catchments have different $x$ ($x_1$, $x_2$, $x_3$, …, $x_N$). The sentence was revised to (**Lines 299-301**): 'The ORCHIDAS with L-BFGS-B algorithm explores the full space of $x$ by perturbing a separate $x$ ($x_i$) over the $i$ th upstream catchment ($i=1, 2, …, N_{opt}$; $N_{opt}$ is the total number of optimized $x$ depending on the number of observation stations) in each iteration'.

**7.** *L274: . . . value '1' and a 'pre-estimated error': 'and' should be 'or'?*

**Answer:** Revised (**Line 310**).

**8.** *L282: the cost function is lower? The value of the cost function? Section Experiments design needs to be a bit clearer.*

**Answer:** 'The cost function is lower …' was revised to 'The value of the cost function …' (**Line 318**).

The following modifications were done in section 'Experiments design' to make it clear.

At **Lines 301-304**: 'To save computing time, the river routing parameterization (forced by corrected $R$ and $D$) rather than the full ORCHIDEE is executed. The total execution time depends on the number of parameters to be optimized, the length of simulation years, and the number of iterations'.

At **Lines 307-308**: 'Over the Iberian Peninsula, the range of $x$ is defined between 0 and 20 which depends on the $Q_{fg}$ and $Q_{obs}$.'

At **Section 2.4** and **Fig. 3**, the expressions of '$x_{prior}$ is set to a constant value 1' and '$x_{prior}$ is set to pre-estimated-prior' are names as $x_{prior\_1}$ and $x_{prior\_ref}$, respectively.

At **Lines 321-324**: 'The oscillation of $J$ at the steps 3 and 5 could be due to the fact that the calculation of the gradient of $J$ by finite difference is not optimal. It is also possible because the L-BFGS-B explores partly the physical range during the first few iteration to estimate the Hessian of the cost function for convergence'.

At **Lines 351-355**, the Eq. (10) for 'Uncertainty' and its explanations are moved to **Section 2.4.**

At **Lines 270-271,** explanations added: 'For each GRDC station, the corresponding catchment surface in the model is estimated.'

**9.** *L288: factor m corresponds to number of GRDC stations?*

**Answer:** Yes. Explanations were added (**Line 327**): '(i.e., the number of GRDC station)'.

**10.** *L304: The river routing model runs at each grid cell? The distributed nature of the river routing model is not quite clear.*

**Answer:** No, the river routing model runs over all model grid cells together in each run. More explanations for Y1SP0 were given at **Lines 344-348**: 'Take the Y1SP0 for example, in each iteration, the correction factor $x$ is perturbed by $m$ times. For each perturbation, the ORCHIDEE river routing model runs once with one $x$ (e.g., $x_i$ at the $i$th sub-catchment) being perturbed while the $x$ of other sub-catchments are kept the same. Therefore, the total number of years required for $m$ stations, $n$ iterations and $k$ years assimilation is $m \times n \times k$'.

**11.** *L322: higher than a factor of 1.5?*

**Answer:** Revised (**Line 373**).

**12.** *L359: "Summary" seems misnamed for the amount of text following*

**Answer:** The sections 3.2.1 and 3.2.2 (Summary) in the previous manuscript have been merged into section 3.2, thus the title 'Summary' is removed.

**13.** *L369: They most certainly do. . .*

**Answer:** The word 'could' was removed from the previous manuscript (**Line 430**).

**14.** *L375: → can allow, remove 'of'*

**Answer:** Removed (**Line 436**).

**15.** *L377: → patterns, some inaccuracies in this area*

**Answer:** Revised (**Line 438**).

**16.** *L383: Is it also connected to topographic or other land surface features which might be not well presented by the forcing data or the model itself? Just wondering.*

**Answer:** Yes, it is. Revised in **Lines 442-444**: 'Besides the atmospheric forcing, the uncertainties could also origin from boundary condition (e.g., topographic or other land surface features), model parameter, model structure or missing processes'.

**17.** *L475: GREAM → GLEAM*

**Answer:** Revised (**Line 561**).

**18.** *L479: references, also maybe mention more global attempts to create gridded runoff data? (can be in the introduction).*

**Answer:** Several references were cited at **Lines 570-572**: 'e.g., Boukthir and Barnier, 2000; Mariotti et al., 2002; Struglia et al., 2004; Peucker-Ehrenbrink, 2009; Ludwig et al., 2009; Szczypta et al., 2012'.

A brief review of global gridded runoff data was added in 'Introduction' (**Lines 56-59**): 'Although great efforts have been made for gridded river discharge data at global scale (e.g., RivDIS v1.1, Vorosmarty et al., 1998; Dai and Trenberth, 2002; Fekete et al., 2002), these data are usually at monthly or annual scales and have not been updated with time'. Two new references were added (**Lines 685-687, 841-843**):

Fekete, B. M., C. J. Vorosmarty, W. Grabs.: High-resolution fields of global runoff combining observed river discharge and simulated water balances, Global Biogeochemical Cycles, 16 (3): 15-1 to 15-10, 2002.

Vorosmarty, C. J., Fekete B. M., and Tucker B. A.: Global River Discharge, 1807-1991, V. 1.1 (RivDIS). ORNL DAAC, Oak Ridge, Tennessee, USA. https://doi.org/10.3334/ORNLDAAC/199, 1998.

**19.** *L507: Throughout the paper most errors are attributed to the lack of human influences. For sure other factors also play a large role?*

**Answer:** Yes. The role of other factors were mentioned in **Lines 602-603**: 'The correction factor $x$ can also cover errors in the model structure, model parameter, or boundary conditions (e.g., land surface characteristics imposed to the model)'.

**20.** *Figure3 top: With the logarithmic scaling the lines mostly seem pretty horizontal. Is there a clearly visible gradient when using a different scale? Maybe add this as a window. Missing unit for J?*

**Answer:** A window for iterations 6-15 with normal y-axis and the unit for J (unit: 1) were added on **Fig. 3a**. Its caption was revised to: '
[revised manuscript text omitted]